# Janus electronic state of supported iridium nanoclusters for sustainable alkaline water electrolysis

Yaoda Liu[1], Lei Li[1], Li Wang[2], Na Li[2], Xiaoxu Zhao [3], Ya Chen[1], Thangavel Sakthivel[4] & Zhengfei Dai [1] ✉

Metal-support electronic interactions play crucial roles in triggering the hydrogen spillover (HSo) to boost hydrogen evolution reaction (HER). It requires the supported metal of electron-rich state to facilitate the proton adsorption/spillover. However, this electron-rich metal state contradicts the traditional metal→support electron transfer protocol and is not compatible with the electron-donating oxygen evolution reaction (OER), especially in proton-poor alkaline conditions. Here we profile an Ir/NiPS$_3$ support structure to study the Ir electronic states and performances in HSo/OER-integrated alkaline water electrolysis. The supported Ir is evidenced with Janus electron-rich and electron-poor states at the tip and interface regions to respectively facilitate the HSo and OER processes. Resultantly, the water electrolysis (WE) is efficiently implemented with 1.51 V at 10 mA cm$^{-2}$ for 1000 h in 1 M KOH and 1.44 V in urea-KOH electrolyte. This research clarifies the Janus electronic state as fundamental in rationalizing efficient metal-support WE catalysts.

Hydrogen is acclaimed as an ideal energy carrier in the frame of future carbon-neutral society[1,2]. An eco-friendly and sustainable technique for hydrogen production is electrocatalytic water-splitting while paired with renewable electricity[3,4]. Alkaline water electrolysis (AWE) holds the promise to mitigate the issues of sluggish water oxidation (OER) kinetics and severe catalyst corrosion in acid, enjoying a competitive edge commercially in water electrolyzers[5,6]. However, such an alkaline proton-poor environment will in turn bring an obstacle to the cathodic hydrogen evolution reaction (HER)[7]. Accordingly for the efficient AWE, a prerequisite is to conquer the proton-generation-evolution limitation in alkaline HER but challengeable, calling for innovative strategies and catalyst designs[5,6]. From the phenomenon of thermocatalysis hydrogen overflow, the community has creatively proposed the hydrogen spillover (HSo) effect to boost the HER activity of metal-support (M-S) electrocatalysts, like Pt/TiO$_2$[8], PtIr/CoP[9], etc. Under the HSo perspective, the proton transfer from the metal activator to support can be facilitated by the internal polarization electric field in essence toward the enhanced HER properties[10]. However, it should be noted that previous HSo reports on M-S catalysts were mostly limited to acidic systems, and the HSo-AWE correlation mechanism is rarely involved and unclear in alkaline proton-poor environments.

In principle, the alkaline HER process suffers from the Volmer step with slow kinetics (M + e$^-$ +H$_2$O → M-H$_{ad}$ + OH$^-$, M is the active center, and H$_{ad}$ is the adsorbed H), where proton adsorption and H$_{ad}$ generation should be the premise for HSo (Fig.1a)[10,11]. For the sake of proton adsorption, the supported metal should be in an electron-rich state to facilitate the electrostatic interaction. But this electron-rich metal state will make the HSo-HER process incompatible with the electron-donating OER. Elusively, recent works have obtained both the HER and OER performances in several HSo-effected metal-support nanocatalysts[12]. This contradiction appeals for an innovative revisit into the electronic state of the metal-support nanostructure. In conventional, the community thinks over the metal-support interaction with a typical M → S electron transfer by the interfacial chemical bonding effect (Fig. 1b). However, this M → S transfer will induce an electron (e$^-$)-deficient state of the metal, which is not beneficial for the

[1]State Key Laboratory for Mechanical Behavior of Materials, Xi'an Jiaotong University, Xi'an 710049, P. R. China. [2]State Key Laboratory for Powder Metallurgy, Central South University, Changsha 410083, P. R. China. [3]School of Materials Science and Engineering, Peking University, Beijing 100871, P. R. China. [4]Department of Chemical Engineering, Kumoh National Institute of Technology, Gyeongbuk 39177, South Korea. ✉e-mail: sensdai@mail.xjtu.edu.cn

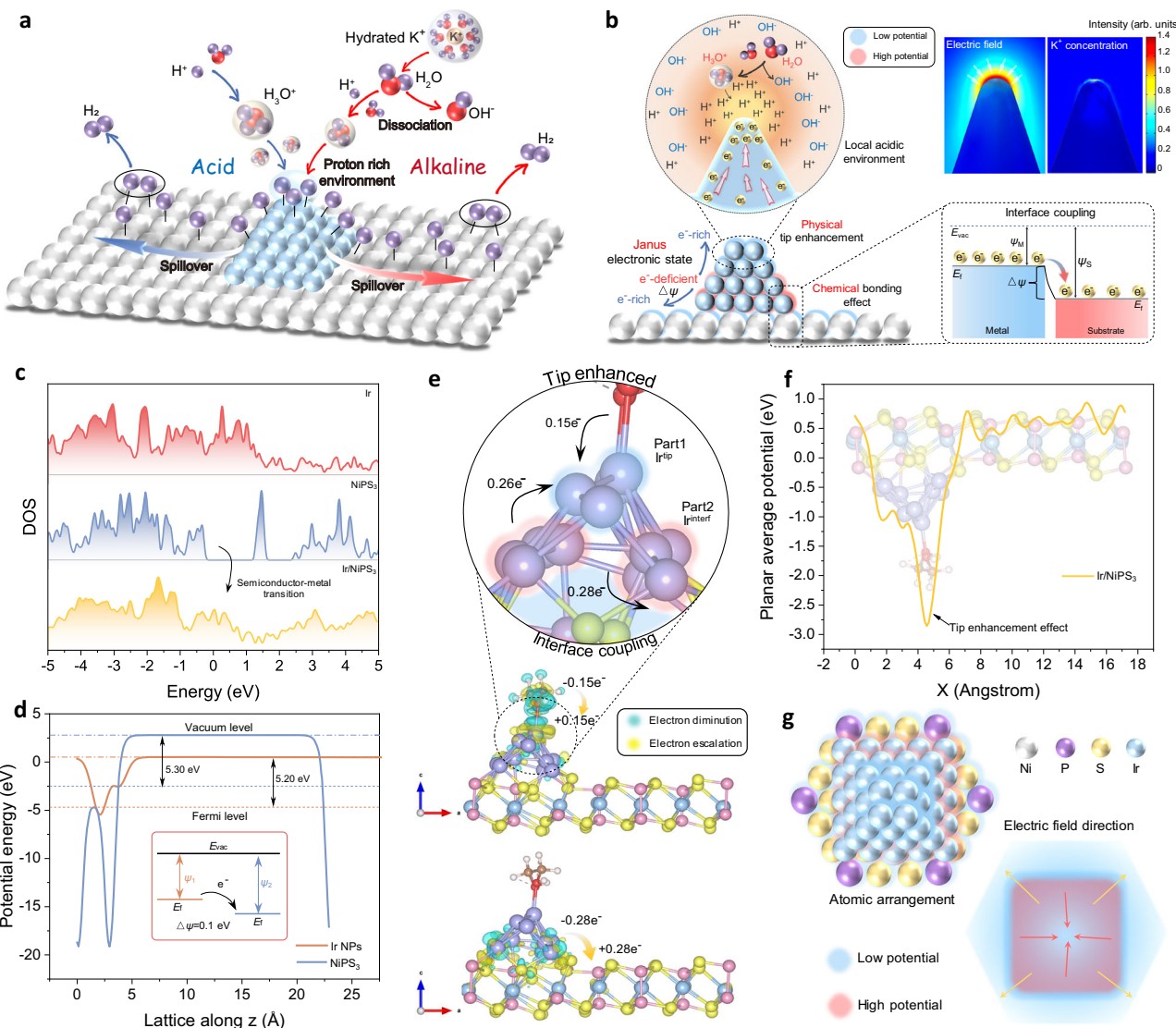

**Fig. 1 | Rationales of the supported metal electrocatalysts for alkaline HSo/ OER-integrated water splitting. a** Comparison of hydrogen spillover under acid (left part) and alkaline (right part) conditions. **b** Physicochemical insight on Janus electronic state in a metal-support structure. The upper right illustration shows the simulation results of tip electric field effect on the cation aggregation. $E_{vac}$, $E_f$, $\psi_M$, $\psi_S$, and $\Delta\psi$ represent vacuum level, Fermi level, work function of metal, work function of substrate, and work function difference, respectively. **c** DOS of Ir, NiPS$_3$, and Ir/NiPS$_3$. **d** Work functions of Ir NPs and NiPS$_3$. **e** The charge density difference between different parts in Ir/NiPS$_3$ by Bader charge analysis. **f** Planar average potential along X direction at the Ir/NiPS$_3$ interface. **g** The top-view scheme of the atomic arrangement and its corresponding electric field direction in Ir/NiPS$_3$.

surface proton adsorption-aggregation in HSo-HER. Recently, oper- ando studies evidenced that proton can accumulate on the supported metal surface in base, creating a local acidic environment to boost HER[13]. This proton accumulation behavior also disaccords with the intrinsic e⁻-deficient state of supported metal. Hence, the conventional M → S one-way electron transfer model would be no longer qualified for rationalizing the supported metal electrocatalysts with HSo effect, and a substantial physicochemical paradigm is extremely desirable.

For a typical metal-support stacking, the electron will be accu- mulated at the top region in the metal due to the physical tip- enhancement effect (Fig. 1b). On account of this, it will form a Janus electronic state in the interfacial (e⁻-deficient) and top (e⁻-rich) regions of the supported metal, empowering a synergy of the alkaline HSo-HER and OER processes. To elucidate the hypothesis, we construct an Ir/ NiPS$_3$ metal-support heterostructure as an electrocatalytic platform for performance and mechanism studies. The Janus electronic state of the supported Ir clusters is theoretically profiled and verified by dif- ferential phase contrast-scanning transmission electron microscopy

(DPC-STEM). In-situ observations indicate the local acidic environment and the dynamic Ir→NiPS$_3$ hydrogen spillover on the catalytic surface for alkaline HER, and the Ir/NiPS$_3$ structure is found with the anti- reconstruction merit in OER. Resultantly, the overall water splitting is sustainably implemented with low cell voltages of 1.51 V in 1 M KOH for 1000 h and 1.44 V in urea-KOH electrolyte at 10 mA cm⁻². This research clarifies the Janus metal electronic state as the fundamental perspec- tive in synergizing the hydrogen spillover and OER toward efficient metal-support AWE catalysts.

## Results

### Theoretical viewpoint for Janus electronic states
The electronic state of Ir/NiPS$_3$ support structure was theoretically profiled by density functional theory (DFT) calculations, together with its effect on HSo-HER/OER processes. The atomic models of NiPS$_3$, Ir surface, and Ir/NiPS$_3$ are presented in Supplementary Fig. 1. Since the ethylene glycol (EG) ligand is used for the later Ir synthesis, the ligand was also taken into account here to construct a more exact Ir/NiPS$_3$

model[14]. Figure 1c compares the density of states (DOS) of $NiPS_3$, Ir, and $Ir/NiPS_3$. Upon stacking Ir on the semiconducting $NiPS_3$, the heterostructure exhibits a metallic characteristic that would facilitate the electron transfer in electrochemical catalysis. Besides, the work function difference ($\Delta\psi$) between Ir nanoparticle and $NiPS_3$ support was evaluated in Fig. 1d. It indicates an overall electron transfer from Ir to $NiPS_3$, and records a small $\Delta\psi$ of 0.1 eV between them. Such a small $\Delta\psi$ barrier will weaken the charge accumulation and proton capture at the interface to rationalize the hydrogen spillover[9].

Figure 1e presents the charge density difference and Bader charges analysis to further dig out the accurate charge distribution of the system. The Ir atoms at the interface ($Ir^{interf}$) are revealed to donate electrons to both the $NiPS_3$ support and tip-region Ir atoms ($Ir^{tip}$). This bidirectional electron transfer manner induces the formation of $e^-$-deficient $Ir^{interf}$ region and $e^-$-rich $Ir^{tip}$ region, verifying the hypothesis of Janus electronic state in Fig. 1b. In addition, the EG ligand supplies electrons to $Ir^{tip}$ to consolidate this Janus electronic state, which is beneficial for proton enrichment at the tip. We further indicate the plane average potential along the X direction for the $Ir/NiPS_3$ heterostructure (Fig. 1f), evidencing a sharp increase in electrons at the Ir tip. The $Ir^{tip}$ region with enriched electrons can activate a local acidic proton-rich environment for alkaline hydrogen spillover[13]; while the $e^-$-deficient $Ir^{interf}$ region will take effect in the electron-donating OER processes[15]. Hence, this Janus electronic state is in line with the demand for the HSo-HER/OER integrated water electrolysis. Based on the above analyses, an idealized top view of the potential distribution and electric field direction is presented in Fig. 1g for the $Ir/NiPS_3$ heterostructure. This non-uniform field flow and electron configuration will be specifically verified through the experiments later.

## Morphologies and microstructures

To examine the hypothesis, we have loaded Ir nanoparticles (NPs) with $NiPS_3$ lamella by an in-situ EG reduction method. The synthesis of $Ir/NiPS_3$ heterostructures schemes in Supplementary Fig. 2. It starts from the electrochemical exfoliation of layered bulk $NiPS_3$ crystal (Supplementary Fig. 3a) to few-layer $NiPS_3$ (EE-$NiPS_3$, Supplementary Fig. 3b)[10,16]. The dimensional statistics (Supplementary Fig. 4) indicate that EE-$NiPS_3$ is of ultra-thin nanosheet with a thickness of ~2.60 nm (ca. 4 layers). Finally, by EG reduction process, Ir NPs can be loaded on EE-$NiPS_3$ to form $Ir/NiPS_3$ metal-support structure (Supplementary Fig. 5). Pure Ir NPs (Supplementary Fig. 6) were also synthesized for reference.

The aberration-corrected transmission electron microscopy (AC-TEM) and high-angle annular dark-field scanning TEM (HAADF-STEM) images (Fig. 2a–c) clearly show that the size of the loaded Ir NPs is uniform with an average particle size of ~1.94 nm (Fig. 2b, inset). The ultrafine grain distribution may provide more exposed catalytic centers[14,17]. The magnified HAADF-STEM image clearly illustrates the crystallinity of Ir NPs with legible lattices (Fig. 2c). By focusing on a single NP, lattice fringes can be observed and ascribed to the Ir (111) crystal plane (Fig. 2c, inset, upper right). The 2D planar image based on thickness-contrast simulations (Fig. 2c inset, bottom left) provides a more intuitive illustration of the characteristic morphology. It shows that the center of Ir NP is higher and narrower than its edge region, resembling a pyramid-like structure. A more intuitive high-resolution TEM image (Fig. 2d) shows the d-spacing of 0.289 nm (130) of $NiPS_3$ and 0.235 nm (111) of Ir, demonstrating the growth of Ir particles on $NiPS_3$. In the selected area electron diffraction pattern (SAED, Fig. 2e), the $NiPS_3$ (110)/(130)/(131)/(133)/(352) planes of and Ir (111) plane can be clearly identified in the heterostructure. The successful incorporation of Ir NPs with $NiPS_3$ was further confirmed by the elemental mapping (Fig. 2f), where the distribution of each element was relatively uniform.

## Phase, surface states, and coordination environment

Supplementary Fig. 7 shows the X-ray diffraction (XRD) patterns of bulk $NiPS_3$, EE-$NiPS_3$, and $Ir/NiPS_3$ heterostructures. For $Ir/NiPS_3$, the peaks at 14.1°/36.0°/49.5°/54.5°/57.8° respectively correspond to the (001)/(131)/(202)/(060)/(133) planes of $NiPS_3$, and no Ir peak was found due to the detection difficulty of ultrafine NPs (<2 nm) by XRD diffractometer[18]. Supplementary Fig. 8 presents their Raman spectra. The peaks of $Ir/NiPS_3$ heterostructures undergo a red Raman shift relative to EE-$NiPS_3$. This is attributed to the hindered vibration activity of the $NiPS_3$ substrate atoms after Ir loading[19], suggesting an intimate interaction between Ir and $NiPS_3$. From the Fourier transform infrared spectra (FT-IR, Supplementary Fig. 9), the $Ir/NiPS_3$ heterostructure has a stronger interaction with water than $NiPS_3$, because of the higher O-H vibration intensity. Besides, the FTIR peaks of $Ir/NiPS_3$ in $-CH_2$, C-H, and C = O regions are consistent with EG, indicating the presence of EG ligands on the surface. The valence states were further examined using X-ray photoelectron spectroscopy (XPS) in Fig. 3a, b and Supplementary Fig. 10–12. Relative to EE-$NiPS_3$, the Ni $2p$ peaks shift about −0.29 eV for $Ir/NiPS_3$ material (Fig. 3a), indicating the electron acceptor role of $NiPS_3$ here[20]. Ir $4f$ spectra of Ir NPs and $Ir/NiPS_3$ samples were compared in Fig. 3b, showing the co-existence of $Ir^0$ and $Ir^{4+}$ doublets[17]. The $Ir^{4+}/Ir^0$ ratios were further calculated to 0.61/1 for Ir NPs and 0.64/1 for $Ir/NiPS_3$. From the whole perspective, the Ir valence increase also indicates the electron donation from Ir NPs to $NiPS_3$ in the heterostructure.

Figure 3c presents the Ni K-edge X-ray absorption near edge structure (XANES) spectra of $Ir/NiPS_3$. Compared to EE-$NiPS_3$, the downshift of XANES edge presents the reduction state of Ni in $Ir/NiPS_3$ (Fig. 3c, inset)[10]. While in the Ir $L_3$-edge XANES spectra, the higher white line intensity manifests the reduced electronic density and the increased Ir oxidation state in $Ir/NiPS_3$ (Fig. 3d, inset)[17]. The XANES results are consistent with those of XPS and DFT data, indicating an electron transfer from Ir NPs to $NiPS_3$. The local coordination environment was further studied in Fig. 3e by the Fourier transform function extended X-ray absorption fine structure (FT-EXAFS) spectra with phase correction. Compared with EE-$NiPS_3$, characteristic peaks representing the Ni-S and Ni-Ni coordination shifted significantly to a low R-value in $Ir/NiPS_3$. This reflects that the growth of Ir NPs on $NiPS_3$ induces structural distortion of the $NiPS_3$ base plane[10]. As for Ir environment (Fig. 3f and Supplementary Fig. 13, 14), the Ir-Ir bond length in the $Ir/NiPS_3$ is slightly longer, implying a tensile stress in Ir NPs by the substrate confinement. Wavelet transforms (WT) for the $k^3$-weighted EXAFS signals further demonstrate the changes in the coordination environment and bond length (Fig. 3g–l). Relative to EE-$NiPS_3$ (Fig. 3g), Ni-S and Ni-Ni bonds in $Ir/NiPS_3$ have smaller R values with lower intensities (Fig. 3h), indicating the shrinkage of coordination number and bond length. Accordingly, Ir in the heterostructure shows a new coordination environment for Ir-Ni and a larger R-value for Ir-Ir coordination (Fig. 3j–l). The XAFS results clearly state that Ir NPs undergo chemical bonding with the $NiPS_3$ support, resulting in a distortion of valence and coordination environments.

## Catalytic properties for water electrolysis

Upon loading on glassy carbon working electrode, the HER properties of different catalysts were firstly evaluated from the polarization curve and Tafel slope in 1 M KOH electrolyte (Fig. 4a, Supplementary Fig. 15a). The $Ir/NiPS_3$ composite exhibits superior HER activity and kinetics with a low $\eta_{10}$ overpotential (at 10 mA cm$^{-2}$) of 23 mV and Tafel slope of 32 mV dec$^{-1}$, beyond those of bulk $NiPS_3$ (409 mV, 102.5 mV dec$^{-1}$), EE-$NiPS_3$ (233 mV, 113.2 mV dec$^{-1}$), Ir NPs (115 mV, 100.6 mV dec$^{-1}$), commercial Ir/C (97 mV, 97.6 mV dec$^{-1}$) and Pt/C (39 mV, 50.8 mV dec$^{-1}$). Besides, the low onset-potential (2.5 mV), $\eta_{10}$, and Tafel slope of this $Ir/NiPS_3$ powder catalyst are comparable with the state-of-the-art HER electrocatalysts (Supplementary Table 1). In the acidic electrolyte, a similar trend can also be observed (Supplementary Fig. 16). Electrochemical active surface area (ECSA) is further applied to evaluate the intrinsic catalytic activity, as estimated by the double-layer capacitance ($C_{dl}$, Supplementary Fig. 17)[21]. Figure 4b displays the higher

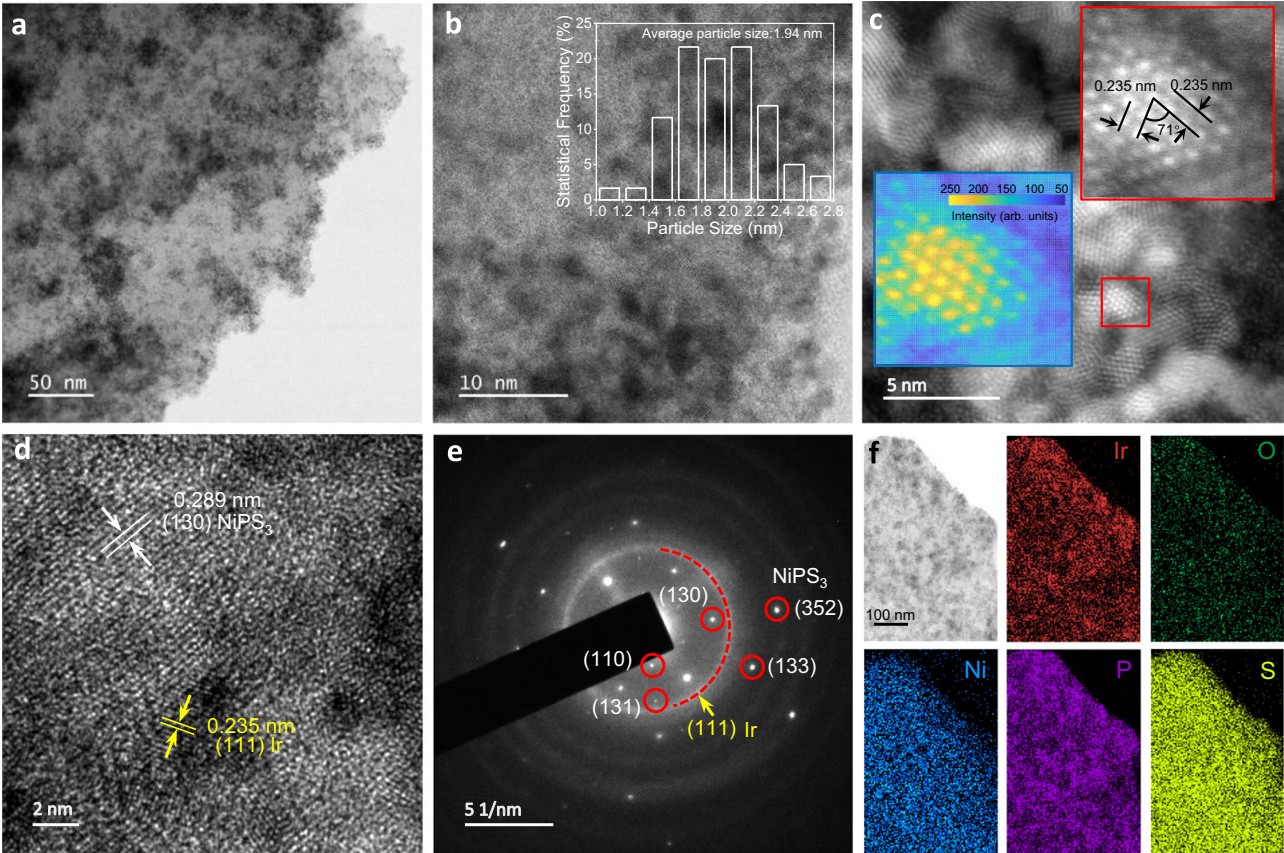

**Fig. 2 | Structural characterizations of the Ir/NiPS₃ heterostructure. a, b** AC-TEM images (inset of b, the average particle size of loaded Ir NPs). **c** HADDF-STEM image, **d** high-resolution TEM image, **e** SAED pattern, **f** elemental mapping image.

$C_{dl}$ and ECSA of Ir/NiPS₃ (35.92 mF cm⁻²) than those of EE NiPS₃ (17.68 mF cm⁻²), Ir/C (21.80 mF cm⁻²), and other samples. Moreover, the Ir/NiPS₃ catalyst performs much better cycling stability within 10000 cycles than commercial Pt/C (Fig. 4c). After cycling test, this Ir/NiPS₃ heterostructure still reserved the particle-on-nanosheet morphology without apparent microstructure and element changes (Supplementary Figs. 18–21).

The OER catalytic properties are also estimated in 1 M KOH electrolyte. Figure 4d and Supplementary Fig. 15b reveal the $\eta_{10}$ and Tafel slope for Ir/NiPS₃ (236 mV, 83.2 mV dec⁻¹). The values are lower than those of EE-NiPS₃ (337 mV, 155.5 mV dec⁻¹), commercial IrO₂ (330 mV, 101.6 mV dec⁻¹), Ir NPs (313 mV, 84.7 mV dec⁻¹), RuO₂ (329 mV, 117.9 mV dec⁻¹), and other samples. Such an improved OER performance of Ir/NiPS₃ is comparable to advanced OER electrocatalysts in Supplementary Table 2. From the electrochemical impedance spectroscopy (EIS) experiment at OER $\eta_{10}$ potentials (Fig. 4e), the lowest charge-transfer resistance ($R_{ct}$) of Ir/NiPS₃ also manifests the fast electron transport characteristics. For OER stability, a 40 h chronopotentiometry measurement was carried out (Supplementary Fig. 22). Unlike IrO₂ and Ir NPs catalysts (quick decay), the Ir/NiPS₃ heterostructures show more practical advantages in OER stability[22]. After OER cycling test, the structural characteristics of Ir/NiPS₃ are presented in Supplementary Fig. 23, 24. And the surface structure of Ir/NiPS₃ heterostructure does not change significantly after OER cycling, showing a certain degree of resistance to surface reconstruction. This structural integrity provides a credible platform for the theoretical modeling and prediction of the OER active centers. The HER/OER mass activities (MA) of these catalysts were compared in Supplementary Fig. 25, where the Ir/NiPS₃ exhibited several times higher MA than commercial electrocatalysts.

Due to the sluggish anodic OER process, water electrolysis always requires a large theoretical overpotential. The urea oxidation reaction (UOR) with lower voltage windows is deemed as an effective way to replace OER toward energy-saving H₂ generation[23]. Therefore, we have also evaluated the two half-reactions of urea-assisted water splitting. As shown in Supplementary Fig. 26, in the electrolyte of 1 M KOH with 0.5 M urea, the HER activity of Ir/NiPS₃ catalyst did not change significantly. While serving at the anodic reaction, each catalyst needs a lower potential for UOR than OER (Fig.4f, Supplementary Fig. 27). Among these samples, Ir/NiPS₃ also shows the most active alkaline UOR ability in terms of potential (1.36 V vs. RHE at 10 mA cm⁻²) and Tafel slope (21.1 mV dec⁻¹), superior to many reported UOR catalysts (Supplementary Table 3). In addition, the Ir/NiPS₃ performs stable in UOR electrocatalysis (Supplementary Fig. 27b), and maintains structural and elemental stability after HER and UOR cycling tests in alkaline urea electrolyte (Supplementary Figs. 28–31).

For overall water splitting (OWS), the Ir/NiPS₃ || Ir/NiPS₃ electrode couple demonstrates a lower $\eta_{10}$ cell voltage of 1.51 V (Fig. 4g) than the Pt/C || RuO₂ couple (1.59 V). A lower $\eta_{10}$ of 1.44 V was further achieved with the Ir/NiPS₃ catalyst in urea-assisted OWS (UA-OWS, Fig. 4g inset) with good durability (Supplementary Fig. 32). The OWS and UA-OWS performances are comparable to many current advanced catalytic systems (Supplementary Table 4, 5)[24–27]. It is also found that the catalytic electrode pair could remain stable for over 1000 h at varied OWS current densities (Fig. 4h). Figure 4i clearly demonstrates the potential of Ir/NiPS₃ to compete with the current state-of-the-art catalysts in OWS and UA-OWS. A wind power-driven UA-OWS device was also designed in Fig. 4j to reflect the capacity in sustainable hydrogen production. In the wind-self-powered (-1.44 V) electrolyzer, the bubbles can be clearly observed on the electrode surface. Specifically, such

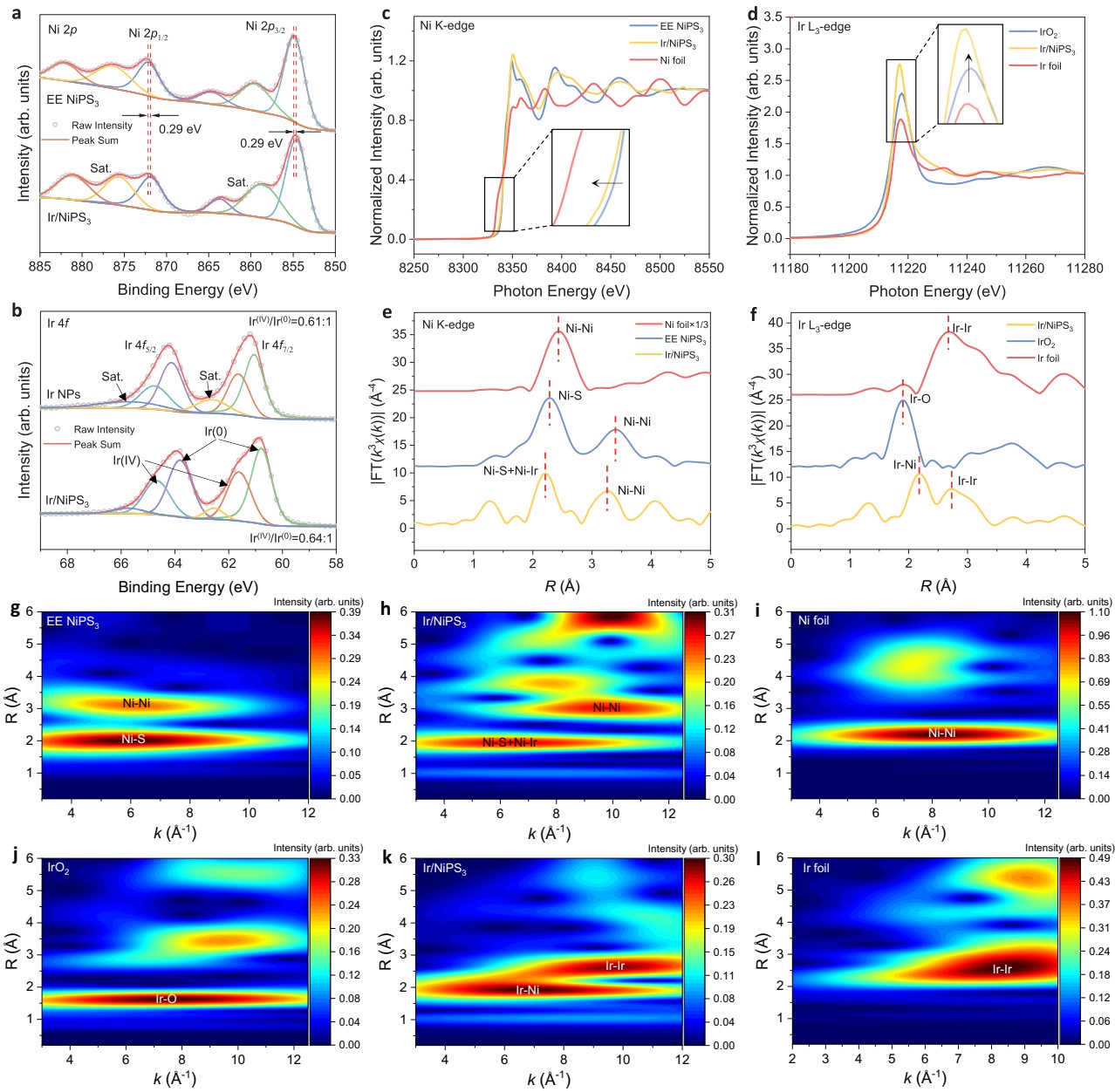

**Fig. 3 | Electronic and coordination structure of Ir NPs, NiPS₃ and Ir/NiPS₃. a** Ni 2$p$ XPS spectra of EE NiPS₃ and Ir/NiPS₃. **b** Ir 4$f$ XPS spectra of Ir NPs and Ir/NiPS₃. **c** Ni K-edge XANES of EE NiPS₃, Ir/NiPS₃, and Ni foil. **d** Ir L₃-edge XANES of IrO₂, Ir/NiPS₃, and Ir foil. **e** Ni K-edge FT-EXAFS of EE-NiPS₃, Ir/NiPS₃, and Ni foil. **f** Ir L₃-edge FT-EXAFS of IrO₂, Ir/NiPS₃, and Ir foil. WT for the $k^3$-weighted EXAFS signal of **g** Ni K-edge in EE-NiPS₃, **h** Ni K-edge in Ir/NiPS₃, **i** Ni K-edge in Ni foil, **j** Ir L₃-edge in IrO₂, **k** Ir L₃-edge in Ir/NiPS₃, and **l** Ir L₃-edge in Ir foil.

a UA-OWS device can be also applied to the urea-contaminated water degradation (Fig. 4k and Supplementary Fig. 33). By comparison with standard colorimetric cards, a reduction in urea content in contaminated water was visually demonstrated, which can meet the water standards in some scenarios like swimming pools[28]. Besides, the contact angle measurement in Supplementary Fig. 34 further confirms the stronger hydrophilicity of the heterostructure for better water activation on the catalytic surface[10]. The above results indicate that the Ir/NiPS₃ metal-support catalyst can be actively performed in sustainable water electrolysis applications.

### Detailed analyses for the enhanced catalytic mechanism

As proposed in Fig. 1, the Janus electronic state of the support Ir could synergize the alkaline hydrogen spillover HER and OER processes. To gain the experimental evidence, we measured local electric fields on the surface of Ir/NiPS₃ catalysts using DPC-STEM technology[29]. The color plot in Fig. 5a shows the non-uniform distribution of the electric field (EF) around a single Ir NP on NiPS₃. The arrow presents the irregular direction of electron transfer around the Ir particle, indicating that the electric field around Ir is anisotropic. The EF at the Ir atom near the edge of the substrate shows a direction to the substrate; while inside the Ir NP, the EF direction is oriented to the center Ir atoms. The in-plane overlapped field flow magnitude has been intuitively demonstrated with Janus electronic state (Fig. 5b), in accordance with the theoretical prediction result (Fig. 1g). The bidirectional electron transfer and Janus electronic state were also observed on many other selected Ir NPs in the heterostructure (Supplementary Fig. 35). Further, surface potentials were estimated using Kelvin Probe Force Microscopy (KPFM) on a more macroscopic scale. The Ir/NiPS₃ sample showed a greater

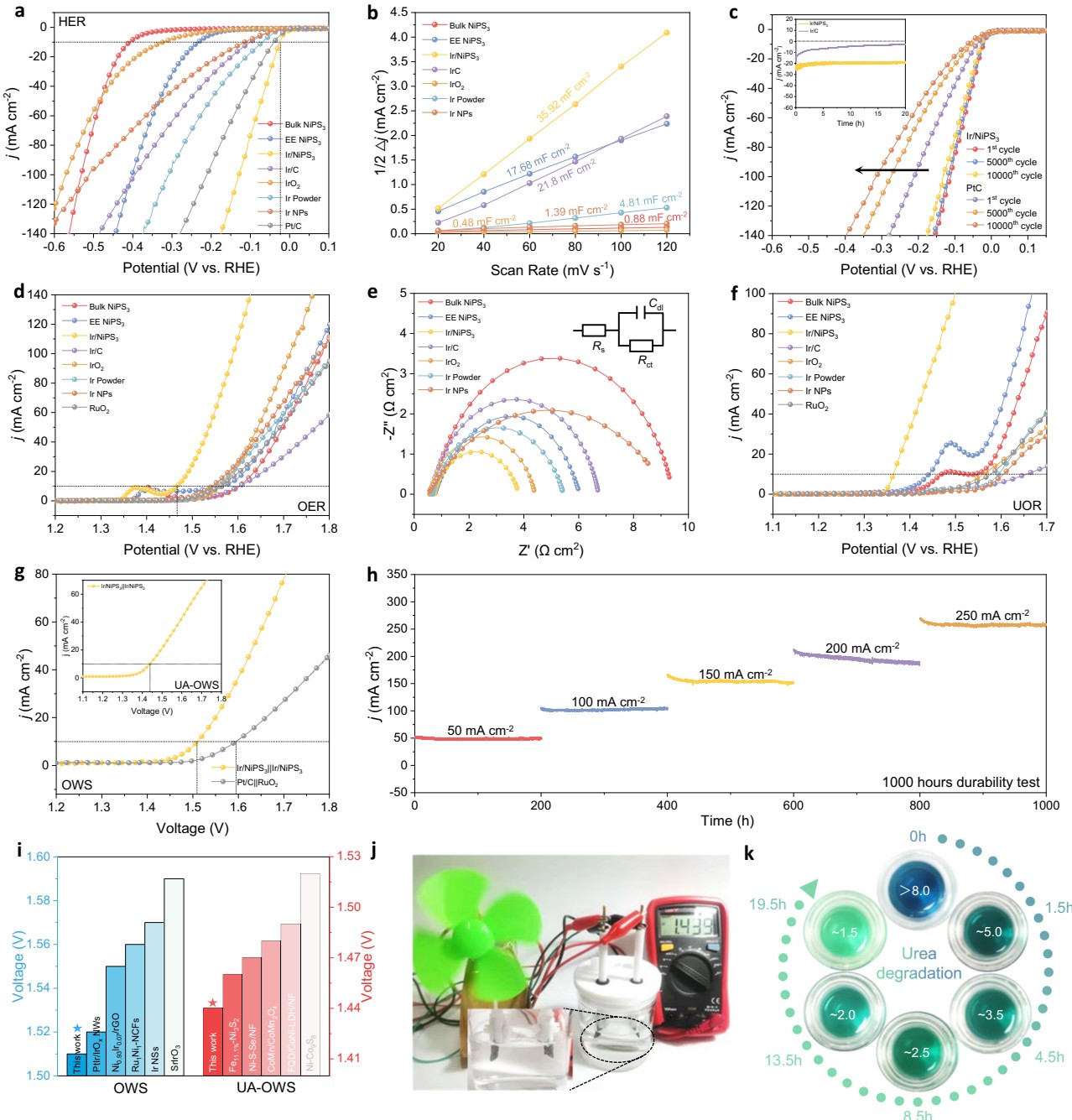

**Fig. 4 | Electrocatalytic performance studies of different catalysts. a** LSV curves, **b** Current-potential plots at different scan rates, and **c** Cycling durability test curves in 1 M KOH for HER. **d** LSV curves and **e** EIS data in 1 M KOH for OER. **f** UOR LSV polarization curves in 1 M KOH + 0.5 M urea. **g** Polarization curves of OWS based on Ir/NiPS$_3$||Ir/NiPS$_3$ couple in 1 M KOH (inset, in 1 M KOH + 0.5 M urea). **h** Long-term OWS chronopotentiometry measurement of Ir/NiPS$_3$||Ir/NiPS$_3$ couple for 1000 h. **i** Comparison of the OWS and UA-OWS $\eta_{10}$ cell voltage of Ir/NiPS$_3$ with reported catalysts. **j** Photograph of the wind-powered UA-OWS device. **k** Urea contaminated water degradation by UA-OWS using EasyBox detection reagent. Supplementary Fig. 33 provides the standard colorimetric card.

surface potential ($\Delta E_s$ = 34.2 mV, Fig. 5c, d) compared with pure NiPS$_3$ ($\Delta E_s$ = 6.9 mV, Supplementary Fig. 36). The enhancement of the surface potential fluctuation greatly affects the charge redistribution, which will regulate the selective intermediates adsorption/desorption in different regions for HER and OER. Note that the electron-poor region at the interface is conducive to the aggregation of electron-rich OH$^-$ for UOR dehydrogenation[30].

The structural evolution during the actual electrocatalysis was monitored using in-situ Raman spectroscopy (Fig. 5e–g). For alkaline HER, the Raman spectra of Ir/NiPS$_3$ and NiPS$_3$ were measured at open circuit, −50 ~ −150 mV vs. RHE, respectively (Fig. 5e). As the applied

potential gradually decreases, a Raman peak newly appears at ~1750 cm$^{-1}$ in Ir/NiPS$_3$, belonging to the H$_3$O$^+$ intermediate species[13,31]. This indicates that the loading of Ir NPs contributes to generating H$_3$O$^+$ species and forming the local acidic environments for alkaline hydrogen spillover[13]. Note that the $E_g^{(2)}$ in-plane tensile vibration mode of NiPS$_3$ (ca. 177.1 cm$^{-1}$, Supplementary Fig. 37a) is very sensitive to changes in the surface structure of NiPS$_3$[32]. As for the Ir/NiPS$_3$ sample (Fig. 5f), the peak of $E_g^{(2)}$ undergoes a blue shift during the HER when a negative potential was applied, and it is ever-shifting as the catalytic reaction progress. This Raman blue shift manner is not observed in the pure NiPS$_3$ surface during the HER process (Supplementary Fig. 37b).

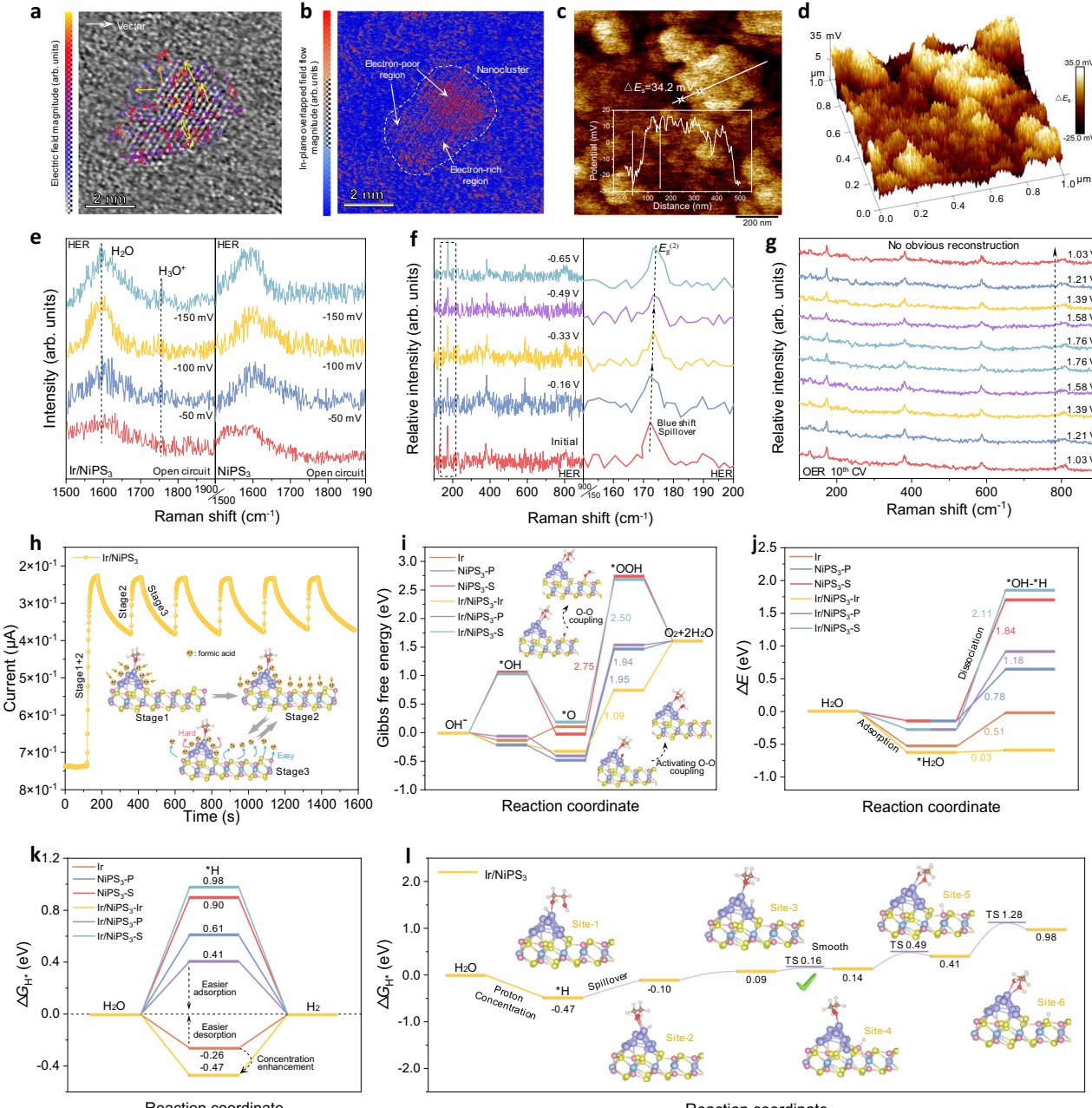

**Fig. 5 | Experimental and theoretical insights for water electrolysis mechanisms. a** DPC-STEM images of the Ir/NiPS$_3$ catalyst, and **b** the corresponding plot of the in-plane overlapped field flow magnitude. **c** Surface potentials of Ir/NiPS$_3$ catalyst measured by KPFM and **d** its corresponding 3D diagram. **e** The in-situ Raman spectra of Ir/NiPS$_3$ (Left) and NiPS$_3$ (Right) at different applied potentials vs. RHE. **f** The in-situ Raman spectra of Ir/NiPS$_3$ at higher applied potentials for HER. **g** The in-situ Raman spectra of Ir/NiPS$_3$ during OER cycling. **h** Sensing response of Ir/NiPS$_3$ when cycled between dry air and formic acid-mixed air. **i** Energy barriers for four-electron-step OER process. **j** Energy barriers for water adsorption and dissociation. **k** Energy barriers for HER on different sites for Ir, Ir/NiPS$_3$, and NiPS$_3$. **l** Calculated free energy diagram for hydrogen spillover on Ir/NiPS$_3$.

According to previous protocols[8], it is reasonable to speculate that this blue shift may be due to hydrogen spillover from Ir NPs to NiPS$_3$. Moreover, Supplementary Figs. 38–44 provide the additional evidence for alkaline hydrogen spillover and the promoted water dissociation kinetic by the loaded Ir[9,33–36], details are presented in the Supplementary Information. With respect to OER, we selected the 10th CV scanning cycle to analyze the in-situ Raman spectrum (Fig. 5g, Supplementary Fig. 45). No significant new peaks were observed during the OER potential change, indicating that NiOOH was not formed within a certain operating period. This result once again verifies that Ir/NiPS$_3$ has a certain resistance to OER surface reconstruction, and the stable two-phase interface is crucial for accurate mechanism analysis.

In order to understand the enhanced proton concentration feature of Ir/NiPS$_3$, we innovatively combined the gas sensing test to provide an alternative measure for the proton affinity of catalysts (Fig. 5h and Supplementary Fig. 46)[10]. Formic acid was used as a typical proton solvent in the gas sensing tests. The EE-NiPS$_3$ sample exhibited rapid response/recovery to formic acid vapor, indicating the easy proton adsorption/desorption (Supplementary Fig. 46b); whereas the pure Ir NPs showed a strong proton capture effect without sensing recovery (Supplementary Fig. 46c). Particularly, the response/recovery process of Ir/NiPS$_3$ to formic acid vapor is manifested in three stages (Fig. 5h). It is reasonable to speculate that the proton preferentially adsorbs on the supported Ir NPs during the

first response (Stage 1), and then transfer to the easily desorbed NiPS$_3$ support to conduct the recovery in the following process (Stage 2–3). This result is also consistent with our assumption of the HSo-HER process.

Thermodynamic analyses have been further carried out for detailing the enhanced catalytic mechanism. The adsorbate evolution mechanism (AEM) was applied for the alkaline OER process (Supplementary Note 1, Supplementary Figs. 47–52). Supplementary Table 6 lists the theoretical OER overpotential ($\eta$). From the step energy diagrams in Fig. 5i, the third OER step (*O + OH$^-$ ⟶ *OOH+e$^-$) is the rate-determining step for these electrocatalysts. Throughout the OER process, the edge Ir sites (Ir$^{interf}$) adjacent to the substrate in Ir/NiPS$_3$ exhibit a more balanced energy barrier relative to other sites and materials, due to their e$^-$-deficient state. The electron-rich state of the Ir$^{tip}$ site makes it difficult for OER intermediates to adsorb and automatically transfer to Ir$^{interf}$. Given that H$_2$O adsorption is essential for alkaline HER, we calculated the H$_2$O adsorption sites and energy ($\Delta E_{H_2O}$, Fig. 5j, Supplementary Fig. 53, Supplementary Note 2, and Supplementary Table 7). The supported Ir sites show a stronger water adsorption ability with more negative energy value; and meanwhile, they also present the most energetic water dissociation kinetics (0.03 eV, Supplementary Figs. 54, 55, and Supplementary Table 8). Moreover, the $\Delta G_{H^*}$ values of individual sites in different systems are compared in Fig. 5k, Supplementary Figs. 56, 57, and Supplementary Table 9. Relative to NiPS$_3$ (0.61 eV-P site), the $|\Delta G_{H^*}|$ value of P site (0.23 eV) on Ir/NiPS$_3$ is more neutral for balanced hydrogen interactions toward higher HER activity[37]. Besides, the Ir NPs loaded on Ir/NiPS$_3$ exhibit stronger proton aggregation ability (−0.47 eV) than pure Ir surface (−0.26 eV). This may be due to the tensile stress on the Ir NPs after loading (Supplementary Fig. 58), leading to an increase in their d-band centers to make the $\Delta G_{H^*}$ even more negative[38,39]. This enhancement of proton concentration effect benefits the hydrogen spillover mechanism based on multiple sites. Specific reaction site analyses were further conducted to understand the hydrogen spillover in HER of Ir/NiPS$_3$ heterostructure (Fig. 5l). At site 1, $\Delta G_{H^*} < 0$, Ir$^{tip}$ as a proton activator shows the strong H* adsorption behavior; at site 2–6, $\Delta G_{H^*}$ increases gradually for Ir/NiPS$_3$, and its value at the NiPS$_3$ substrate is sufficient to be >0, resulting in the H* desorption tendency on the substrate. Meanwhile, low proton spillover energy barriers (0.07 eV, 0.35 eV, 0.87 eV) were observed at the interface and substrate, which will be easy to overcome according to previous reports[9,40]. From the thermodynamic point, a hydrogen spillover channel can be formed to promote the proton mass transfer and HER kinetics. Moreover, when the simulated size of the Ir NP was enlarged, new "tips" (positions with large curvature) were formed after complete structural relaxation (Supplementary Fig. 59a). Relevant tendencies have been maintained in terms of metallic electronic structure and interface charge transfer (Supplementary Fig. 59b, c) as well as the Janus electronic state (Supplementary Fig. 60) in the expanded system. It also demonstrates the OER-active Ir$^{interf}$ sites (Supplementary Figs. 61–64, Supplementary Table 10) and similar Ir$^{tip}$→Ir$^{interf}$ → NiPS$_3$ hydrogen spillover channels (Supplementary Fig. 65, Supplementary Table 11).

## Discussion

To summarize, we have first theoretically profiled the Ir/NiPS$_3$ metal-support system that successfully shapes the Janus electron-rich/electron-poor region due to the physical tip enhancement and chemical interface bonding effects. The Ir/NiPS$_3$ heterostructure was constructed by growth of Ir nanoparticles on the exfoliated NiPS$_3$ layers, and featured with the Janus electronic state of supported Ir nanoclusters by DPC-STEM observations. As for alkaline HER, the Ir/NiPS$_3$ heterostructure has achieved the Pt-beyond performances with a low $\eta_{10}$ potential of 23 mV and fast kinetics (32 mV dec$^{-1}$). It also exhibits a complective anti-reconstruction OER activity with a low $\eta_{10}$

potential of 236 mV, and further improved by UOR ($\eta_{10}$ = 1.36 V vs. RHE, 21.1 mV dec$^{-1}$). The optimized Ir/NiPS$_3$ composite electrode pairs were served for OWS with low $\eta_{10}$ voltages of 1.44 V and 1.51 V for 1000 h with/without urea assistance. We have also demonstrated a wind-assisted water electrolyzer for sustainable green hydrogen production and urea degradation. In-situ Raman observations and sensing tests have confirmed the local acidic environment, hydrogen spillover phenomenon, and tip enhancement effect for water adsorption and splitting. The HSo/OER-integrated catalytic mechanism of Ir/NiPS$_3$ heterostructure is clearly profiled by DFT calculations with detailed multisite analyses and more balanced step energy barriers. Our study puts forward the Janus electronic state regulation strategy for the design of efficient metal-support AWE catalysts.

## Methods

### Preparation of electrochemical exfoliated NiPS$_3$ (EE-NiPS$_3$)

The synthesis of bulk NiPS$_3$ crystals used the typical chemical vapor transport technology[41]. The mixture of stoichiometric high-purity Ni (nickel powder), P (phosphorus block, red), S (sulfur block) (Ni/P/S = 1:1:3), and iodine (10 mg cm$^{-3}$) as transport agents was sealed in an evacuated quartz ampoule and stored in a two-zone furnace (650-600 °C) for 1 week. Large bulk NiPS$_3$ can be harvested at lower temperature zone. After that, the electrochemical workstation (Autolab PGSTAT204) was employed to carry out the electrochemical exfoliation under a two-electrode system, with a Pt wire as the counter electrode and bulk NiPS$_3$ as the working electrode. The electrolyte was prepared by dissolving 0.05 M tetrabutylammonium tetrafluoroborate (TBAB, 98%) in 30 mL N, N-dimethylformamide (DMF, 99.5%). During the exfoliation process, a static bias voltage of −5 V was applied to the working electrode, causing the bulk NiPS$_3$ to expand and peel off, forming a brown suspension. This suspension was then subjected to an ice bath ultrasonic treatment at 300 W power for 90 min to further exfoliate and disperse the NiPS$_3$ layers. After the ultrasonic treatment, the product was washed with ethanol five times to remove any residual electrolytes or impurities. Finally, the product was vacuum-dried at 60 °C overnight.

### Construction of Ir/NiPS$_3$ heterostructure

EE-NiPS$_3$ was used as a substrate for the growth of Ir nanoparticles. Firstly, uniformly disperse 12 mg EE-NiPS$_3$ in 10 mL ethylene glycol through ultrasound. Later, 420 µL aqueous solution containing 0.1 M K$_2$IrCl$_6$ was added to the above suspension and subjected to ultrasonic treatment for 20 min. Then, the flask was placed in the oil bath which was preheated to 120 °C. After fully removing dissolved oxygen, the mixed solution was added to the flask and maintained at 120 °C for 3 h under continuous magnetic stirring and Ar atmosphere. After the product was cooled to room temperature, it was washed with ethanol and deionized water centrifuged multiple times, and dried under vacuum overnight at 60 °C. The preparation of pure Ir nanoparticles is the same as the above steps, except that EE-NiPS$_3$ was not added.

### Materials characterization

A field emission scanning electron microscope (FESEM, FEI Verios460) was employed to capture nanostructure micrographic images. X-ray diffraction (XRD) patterns were collected using a PANalytical X'Pert Pro diffractometer with Cu Kα1 radiation ($\lambda$ = 1.54056 Å). Morphologies, microstructures, and elemental information were examined through transmission electron microscopy (TEM), high-resolution TEM (HR-TEM) images, and energy-dispersive spectroscopic (EDS) elemental mapping on a JEOL JEM-F2100 instrument. Thermo Fisher Spectra 300 microscopy, operating at 300 kV, was used to acquire double spherical aberration-corrected transmission electron microscopy (AC-TEM) images, differential phase contrast-scanning transmission electron microscopy (DPC-STEM) images, and high angle annular dark-field scanning transmission electron microscopy

(HAADF-STEM) images. Functional groups on the samples were analyzed using a Fourier transform infrared spectrometer (FT-IR, IRPrestige-21). Raman spectra were obtained on a LabRAM HR Evolution instrument with a 532 nm laser excitation source. Thermo Fisher Scientific ESCALAB Xi+ was utilized for X-ray photoelectron spectroscopy (XPS) analysis. The extended X-ray absorption fine structure (EXAFS) and X-ray absorption near edge structure (XANES) measurements were carried out with a synchrotron radiation source and analyzed using Athena software[42]. Kelvin probe force microscopy (KPFM) images were acquired using a Bruker ICON instrument. The optical contact angle was determined with Kruss DSA30 equipment.

### Electrochemical measurements

To prepare the catalyst ink, the catalyst powder (4 mg) and carbon black (1 mg) were ground thoroughly and dispersed in a mixed solution of isopropanol (450 μL), DI water (50 μL), and 5 wt% Nafion (20 μL). The catalytic ink (10 μL) was then uniformly deposited and dried on the glassy carbon (GC, 3 mm) working electrode. Autolab PGSTAT204 workstation with a typical three-electrode system, including a graphite rod counter electrode and a reference electrode Ag/AgCl, was employed to evaluate the HER and OER properties of various catalysts. For HER measurements, $H_2SO_4$ (0.5 M) or KOH (1 M) was used as the electrolyte, while KOH (1 M) was used for OER and KOH (1 M) with urea (0.5 M) for UOR. Linear sweep voltammetry (LSV) curves were recorded at a scanning rate of $5\,mV\,s^{-1}$, with potential ranges of −0.6 to −0.1 V vs. RHE for HER and 1.2 to 1.8 V vs. RHE for OER. Before LSV testing, each electrode underwent 20 cycles of cyclic voltammetry (CV) at a scan rate of $100\,mV\,s^{-1}$. The data were further used for the Ohmic drop (iR) correction. Charge transfer resistance ($R_{ct}$) was fitted using electrochemical impedance spectroscopy (EIS) measurements, applying an alternating current (AC) voltage (5 mV amplitude) over a frequency range of 100 kHz to 0.1 Hz. The electrochemical double-layer capacitance ($C_{dl}$) was assessed to determine the electrochemical active surface area (ECSA) based on CV tests at a potential window of 0.1–0.2 V (vs. RHE) and scan rates of 20, 40, 60, 80, 100, and $120\,mV\,s^{-1}$. All tests were repeated 2–4 times to ensure the reproducibility. The overall water-splitting performance was investigated using a two-electrode device in 1 M KOH and 1 M KOH with 0.5 M urea solutions. LSV polarization curves were measured within a range of 1.1 to 1.8 V (vs. RHE) at a scan speed of $5\,mV\,s^{-1}$. Chronopotentiometry was employed to evaluate the long-term stability of HER, OER, and OWS for the samples. The 1000 h chronopotentiometry measurement for OWS was operated at the corresponding potentials of different current densities on the same electrode pair. The in-situ CV and operando EIS measurements were performed at specified scan rates and overpotentials to verify the occurrence of hydrogen spillover and enhanced water dissociation kinetics, respectively.

### Formic acid sensing tests

A commercial Au interdigital sensing electrode (200 μm spacing, 8 pairs) was coated with the catalytic ink. The schematic representation of the dynamic humidity sensing test device can be found in Supplementary Fig. 45. Dry air was served as the carrier gas, while formic acid served as a proton solvent in the experimental reagent. A mass flow controller was employed to regulate various gas flows. Measurements were conducted at 25 °C, with data automatically acquired by the Keithley 2612B system. Resistance change served as the criterion for evaluating sensing performance. The sensor response (S) was determined by the ratio $R_g/R_a$, where $R_a$ and $R_g$ denote the stable resistance values in dry air and mixed air, respectively.

### Theoretical simulation

Density functional theory (DFT) calculations were conducted using the plane-wave pseudopotential approach within the Vienna ab initio simulation software package (VASP)[43]. The Perdew-Burke-Ernzerhof (PBE) functional, under the generalized gradient approximation (GGA)[44], was employed for simulating electron exchange-correlation energy. Projection-augmented wave pseudopotentials were utilized to describe ion-electron interactions. A 420 eV cut-off energy was set for plane-wave expansion, while the total energy convergence criterion for the self-consistent field method was established at $10^{-5}$ eV. All structures were relaxed using the conjugate gradient method until the force component on each atom reached below 0.02 eV Å$^{-1}$. Brillouin zone sampling was performed using Monkhorst-Pack special k-point meshes[45]. The 4 × 4 × 1, 1 × 2 × 1, and 1 × 2 × 1 k-point grids were applied for the Ir(111), NiPS$_3$, and Ir/NiPS$_3$ systems. Moreover, the Grimme scheme DFT-D3 empirical correction was incorporated to account for van der Waals interactions[46]. Over 20 Å vacuum layers were added in the z-axis direction for all models to minimize interaction between adjacent images due to periodicity. Gibbs free energies were corrected at 298.15 K. Charge density difference studies were employed to analyze electron transfer within the system[47]. The clipping image nudged elastic band method (CINEB) was used to calculate energy barriers[48]. For finite element simulation, the Gouy-Chapman model's Nernst-Planck-Poisson calculation was used to elucidate potential charge transfer and storage mechanisms, as well as ion diffusion controlled by the Poisson equation and Nernst-Planck equation. The Electrostatics and Transport of Diluted Species multiphysical fields in COMSOL Multiphysics 5.6 were applied to evaluate the electrochemical behavior of all species. To accurately describe the properties of particle tips, a 0.1 nm tip radius was employed in the constructed model. The hydrated potassium ion radius (0.33 nm) represented the Helmholtz layer thickness[49]. The absolute temperature $T$ was set to 298.15 K, and the diffusion coefficients ($D$) of hydroxide anions and potassium cations in water were approximately taken as $5.30 \times 10^{-9}$ and $2.14 \times 10^{-9}$ m$^2$ s$^{-1}$, respectively[50,51]. Further details can be found in the Supplementary Notes.

## Data availability

The data that support the findings of this study are available within the article and its Supplementary Information files. Source Data are provided with this paper or obtained from Figshare repository at https://doi.org/10.6084/m9.figshare.25108469.

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

## Acknowledgements

This work was jointly supported by the National Natural Science Foundation of China (52371236 (Z.D.)), China Postdoctoral Science Foundation (2019M663698 (T.S.)), and the Fundamental Scientific Research Project of Xi'an Jiaotong University (xzy022022025 (Y.L.)). We thank Miss Liu at Instrument Analysis Center of Xi'an Jiaotong University for her assistance with XPS analysis. This research used the resources of the HPCC platform in Xi'an Jiaotong University.

## Author contributions

Z.D. and Y.L. conceived and initiated the project. Z.D. supervised the project and designed the experiments. Y.L. and T.S. synthesized the catalysts and performed the electrochemical measurements. Y.L., L.W., N.L., X.Z. and Y.C. performed the XRD, SEM, EDS, AFM, and XPS measurements. L.L. performed the TEM measurements and analyses. Y.L. performed atomistic computations and theoretical analyses. Y.L. and Y.C. conducted the EXAFS measurement. Y.L., T.S. and Z.D. wrote the paper. All authors discussed the results and commented on the paper.

## Competing interests

The authors declare no competing interests.
