## [Peer review file · Nature Communications]

Janus electronic state of supported iridium nanoclusters for sustainable alkaline water electrolysisREVIEWER COMMENTS

Reviewer #1 (Remarks to the Author):

The article titled "Janus electronic state of supported iridium nanoclusters for sustainable alkaline water electrolysis" presents a combined experimental and theoretical study of electrocatalysts design for alkaline hydrogen production. The alkaline hydrogen production is one of the most promising technologies for large-scale green hydrogen generation. However, the main challenge is the relatively low performance of the HER electrocatalysts. In this study, the authors use the hydrogen spillover concept to design a novel hybrid material to address this challenge. The authors found that the Ir metal on NiPS₃ support can have excellent electrocatalytic HER and OER performance. The catalytic HER performance can be further improved when the urea oxidation reaction is used to replace the OER. The results are very interesting and exciting.

However, there are several issues that need to be clarified:

- 1) The authors used the Ir metal cluster to build an atomic model. However, the metal cluster with a tip model cannot be supported by the experimental results from HADDA-STEM and high-resolution TEM images. As a result, this atomic model may not be used to represent the characteristics of the Ir metal on NiPS₃.
- 2) Since the (111) d-spacing values can be identified, I think the Ir nanoparticles are mainly exposed by the {111} facet. Given the small size of the Ir nanoparticles, I doubt that the active site may then be the edge of the {111} facets, not the tip. To this end, the authors need to rebuild the atomic model to investigate their electronic properties and interfacial reactions.
- 3) We can't use the work function of the Ir(111) surface to represent that of the nanoparticles. They may be much different.
- 4) When the in-situ Raman spectra were discussed, the authors mentioned that "the peaks of Eg(2) undergoes a blue shift during the HER", and "this blue shift may be due to hydrogen spillover from Ir NPs to NiPS₃". I agree with the authors, the blue shift may be used as an indicative descriptor. However, it is not direct evidence of the hydrogen spillover. This is because the authors cited Ref. 30 to support this key discovery of this project. Ref. 30 didn't clearly mention that the adsorption of H can lead to the blue shift, as the authors suggested in Fig. S37. So, I don't think the in-situ Raman observations have confirmed hydrogen spillover phenomenon and tip enhancement effect for water adsorption and splitting".
- 5) The results from Ref. 30 demonstrated that the consideration of the magnetic structure of NiPS₃ is important. It is unclear whether the authors considered the magnetic structure of NiPS₃ in their DFT computations.
- 6) Fig. 5I shows the free energy diagram to support the hydrogen spillover process. It is not enough. The energy barrier for each step is needed.

Reviewer #2 (Remarks to the Author):

Dai et al. developed a Ir clusters loaded NiPS₃ metal-support structure, and applied it as an efficient bifunctional HER and OER catalyst. In alkaline environments, it shows a low η_{10} potential of 23 mV for HER and excellent OER/UOR potentials. I am very impressive with the finding of Janus electronic state for the supported Ir nanoclusters as well as the roles in activating the hydrogen spillover and water oxidation. In-situ observations indicated the local acidic environment and the dynamic hydrogen spillover in HER, and found the anti-reconstruction merit in OER. The degradation of urea by UOR is also interesting. Overall, this work presents the novel and essential point of Janus electronic state for metal-support structures, and justifies their findings with both theoretical and experimental evidences. Hence, I recommend this paper to be published on Nat. Commun. after carrying out a minor revision to figure out the following points.

1. The wording in some parts is not clear and makes people confused. For example, the description of the ligand effect of ethylene glycol is not detailed enough, whether ethylene glycol is unintentionally introduced or necessary in synthesis, and whether other polyols can synthesize Ir nanoparticle loaded materials. Besides, the influence of ethylene glycol ligands on Janus electronic states needs further elaboration.
2. Have the authors considered the single atom effect, such as loading Ir as a single atom instead of a particle? Current reports on single atom loading of Ir (Nat. Commun., 2023, 14 (1): 2588; Adv. Material, 2021, 33 (8): 2007056.) all demonstrate their unique advantages. Please provide more explanation.
3. In Figure 3d, from the XAFS date, the Ir exhibits a super high valence state that surpasses Ir in IrO₂. How can understand this? Will it also occur in other metal- support structure? If it is, the authors need to check and provide reference.
4. Please comment on the performance comparison or active sites difference compared to some reports of Ir cluster or particles like 10.1016/j.jchem.2023.08.017 or DOI: 10.26599/NRE.2023.9120056.
5. Although this work focuses on the alkaline performance, it still remain some parts lack of detailed interpretation or descriptions. For example, in Figure S16, compared to alkaline environments, the performance of the Ir/NiPS₃ heterostructure on HER in acidic media is slightly worse, lower than commercial Pt/C. How to understand this phenomenon?
6. Transition metal sites are generally considered active sites for catalytic reactions due to the tunability of d electrons. However, in the DFT calculation section, the authors only calculated P, S, Ir sites, but did not consider Ni sites. Please provide a detailed explanation.
7. I still wonder why the adsorption energy of the Ir site in Figure 5k is lower after the formation of heterostructures (from -0.26 eV to -0.47 eV), but the authors indicated the heterostructure is more

favorable in HER. As we know, ΔG closer to 0 corresponds to more favorable HER. How can this confusion be understood?

Reviewer #3 (Remarks to the Author):

The paper by Liu et al. conducted the induction of Janus electronic states in an Ir nanoparticle-loaded NiPS₃ system to enhance alkaline hydrogen evolution and OER/UOR reactions. This study combines experimental characterization, catalytic testing, and DFT/COMSOL calculations to understand the reasons for forming Janus electronic states and the mechanisms that promote reactions. The concept and research content is interesting and well presented. Additionally, Ir/NiPS₃ bifunctional catalysts exhibit impressive performance, such as low 23 mV/236 mV under alkaline conditions η_{10} HER/OER potential, while also possessing UOR urea degradation ability. Therefore, I recommend its publication in Nat. Commun., after the following concerns and remaining issues are addressed appropriately:

1. In Fig. 1, the authors chose to adsorb an ethylene glycol ligand while constructing the model, which raises doubts about whether the emergence of the Janus electronic state is related to the ethylene glycol ligand. In other words, if the system did not contain the ethylene glycol ligand, would the Janus electronic state not exist? This is fundamental to the discussion of the mechanism in the article and is also the starting point for the authors' choice of the ethylene glycol method for synthesizing the material. Please provide a reasonable explanation.
2. The authors need to declare whether the stability of 1000 hours in Figure 4h was tested on the same sample at different current densities, or was the sample replaced midway?
3. Hydrogen spillover itself is an effect under acidic conditions. Since Ir/NiPS₃ has alkaline hydrogen spillover effect, it should also have acidic hydrogen spillover effect. According to Fig. S16 characterized by the authors, the acidic HER performance of the sample does not exceed that of Pt/C catalyst. Why is its performance not outstanding compared with that of alkaline HER?
4. The potential drop usually occurs due to the solution resistance and needs to be eliminated the catalytic properties. Were the potential drops compensated for the electrocatalytic performance testing?
5. The Supplementary Notes points out that $\Delta G(H^*) = \Delta E(H^*) + 0.24\text{eV}$. However, the data in Supplementary Table 9 seems not to be the case. Please check carefully.
6. For OER reaction, the authors seem to have considered the pH value of the reaction (SI, formula 5-8), and the theoretical potential has been subtracted by 0.401 instead of 1.23 V, indicating that the role of pH value has been considered again. Is this method of considering pH reasonable?
7. In Fig. 5k, why do the authors think that the performance of the P and S sites in Ir/NiPS₃ is improved compared to the pure material, while the performance of pure Ir is not as good as the Ir site in Ir/NiPS₃, even though $\Delta G(H^*)$ is closer to 0 in both cases?

[Responses to reviewer 1's comments]

[Comment]. *The article titled "Janus electronic state of supported iridium nanoclusters for sustainable alkaline water electrolysis" presents a combined experimental and theoretical study of electrocatalysts design for alkaline hydrogen production. The alkaline hydrogen production is one of the most promising technologies for large-scale green hydrogen generation. However, the main challenge is the relatively low performance of the HER electrocatalysts. In this study, the authors use the hydrogen spillover concept to design a novel hybrid material to address this challenge. The authors found that the Ir metal on NiPS₃ support can have excellent electrocatalytic HER and OER performance. The catalytic HER performance can be further improved when the urea oxidation reaction is used to replace the OER. The results are very interesting and exciting.*

[Response] We appreciate the recognition of our work and valuable comments from the reviewer. Indeed, the reviewer's advice and recommendations are very prompt and helpful. We revised and enhanced the manuscript according to your directions. Thank you very much.

[Comment 1]. *The authors used the Ir metal cluster to build an atomic model. However, the metal cluster with a tip model cannot be supported by the experimental results from HADDA-STEM and high-resolution TEM images. As a result, this atomic model may not be used to represent the characteristics of the Ir metal on NiPS₃.*

[Response] Thanks very much for your valuable comments. We will address your concerns from the following aspects:

i) According to the thermodynamic rules of particle growth, supported metal nanoparticles generally tend to nucleate at advantageous sites on the support and grow preferentially in certain orientations. Under the influence of the substrate, the morphology of most nanoparticles and clusters evolves into a bottom-wide and top-narrow pyramid-like structure during gradual growth and aggregation processes to achieve the most stable surface Gibbs free energy equilibrium configuration (*Angew. Chem. Int. Ed.* 2020, 59, 2171-2180; *ACS Catal.* 2020, 10, 19, 11011-11045). Therefore, there will inevitably be a certain curvature at the top or corners, as demonstrated in our simulated results in Fig. 1a, b. The electric field concentrates in these areas (*Angew. Chem. Int. Ed.* 2021, 60, 25766-25770), which is a universal phenomenon. It is evident that our Ir metal model represents this characteristic of charge concentration at the top.

ii) Currently, in theoretical simulations of supported metal nanoparticles, especially in the theoretical calculations related to hydrogen spillover, it is widely acknowledged and adopted to

use small pyramid-shaped (tapered) particle models (*Nat. Commun.* 2022, 13, 5382; *Nat. Commun.* 2021, 12, 3502; *Angew.Chem.* 2023, 135, e202217191; *Angew.Chem. Int. Ed.* 2021, 60, 16622-16627; *Energy Environ. Sci.* 2019, 12, 2298, see the Figure below). This model selects small representative units that can effectively interpret and correlate with experimental results, leading to correct conclusions and saving computational effort. Therefore, given the validity of this model, we also employed a similar structure in our manuscript.

iii) In our work, the Janus electronic state and the associated hydrogen spillover effect and performance enhancement are the core viewpoints. Not only theoretical calculations, but various experimental results such as Fig. 5a-h and the newly added Supplementary Fig. 38-44 have confirmed this viewpoint. Furthermore, the computational results are in agreement with the experimental observations. This indirectly validates the rationality of the theoretical model.

Based on your advice, we have updated the HADDF-STEM image in Fig. 2c. Furthermore, we have incorporated thickness contrast simulations of the nanoparticle to provide a more intuitive 2D planar representation (the left inset in Fig.2c). The brighter central region in the image indicates a greater thickness contrast, thereby demonstrating the characteristic morphology of a higher center and lower edges in the nanoparticle, which is consistent with our theoretical model. In summary, the theoretical model in our work is reasonable and can effectively explain and illustrate the experimental conclusions.

<After revision>

...can be observed and ascribed to the Ir (111) crystal plane (Fig. 2c, inset, upper right). The 2D planar image based on thickness-contrast simulations (Fig. 2c inset, bottom left) provides a more intuitive illustration of the characteristic morphology. It shows that the center of Ir NP is higher and narrower than its edge region, resembling a pyramid-like structure. A more intuitive high-resolution TEM... (1st paragraph, Page 6).

[Comment 2]. Since the (111) d-spacing values can be identified, I think the Ir nanoparticles are mainly exposed by the {111} facet. Given the small size of the Ir nanoparticles, I doubt that the active site may then be the edge of the {111} facets, not the tip. To this end, the authors need to rebuild the atomic model to investigate their electronic properties and interfacial reactions.

[Response] Thanks for your valuable comments. As mentioned in response to the Comment 1, our approach to model construction is actually a combination of experimental evidence and draws inspiration from other references (*Nat. Commun.* 2021, 12, 3502; *Angew. Chem.* 2023, 135, e202217191). And the nanoparticles (NPs) have a thermodynamically stable pyramid-like stacking pattern (Fig. 2c).

Regarding your concerns, we would like to indicate the more detailed information for the model construction (**Fig. R1**). At first, we did choose to expose the (111) crystal plane of Ir (Fig. R1a). Subsequently, the representative Ir unit was loaded on the surface of NiPS₃ for the heterostructure modeling (Fig. R1b). Finally, the Ir/NiPS₃ turns to be a more stable model after structural complete relaxation for reliable calculations (Fig. R1c). The support {111} Ir NP can be divided into two parts, namely the top part (tip, Ir^{tip}) and the bottom edge (Ir^{intef}, connected with the NiPS₃) as you mentioned. Both are indispensable key components for the generation of Janus electronic states. According to the calculation results, this Ir^{intef} edge serves as an electron-poor center to promote OER, while in HER it is a transition site for proton transport.

Fig. R1. The schematic diagram for Ir/NiPS₃ structural modeling process in the manuscript.

To explain your comment more clearly, we have also established a more macroscopic model using larger nanoparticle for reference. As shown in **Fig. R2**, after structural relaxation, the Ir NP loaded on the substrate forms a stacked structure with a narrow top and a wide bottom thermodynamically. The edges of the (111) plane evolve into new generalized “tips” (sites with high curvature inside the particles) and Ir^{interf} at the interface (Supplementary Fig. 59a). The calculation of electronic properties and interface reactions further confirms the charge concentration at these generalized tips, confirming the Janus electronic states (Supplementary Fig. 59,60). The tendency in terms of metallic electronic structure and interface charge transfer has been maintained in the larger system, and it also indicates that the OER-active Ir^{interf} sites and similar Ir^{tip}→Ir^{interf}→NiPS₃ hydrogen spillover channels (Supplementary Fig. 61-65). In summary, the new results are consistent with our previous conclusions. The above results have been also added to the revised manuscript to enrich the content.

Fig. R2. The schematic diagram for an expanded Ir/NiPS₃ structure in the revised manuscript.

<After revision>

...From the thermodynamic point, a hydrogen spillover channel can be formed to promote the proton mass transfer and HER kinetics. Moreover, when the simulated size of the Ir NP was

enlarged, new "tips" (positions with large curvature) were formed after complete structural relaxation (Supplementary Fig. 59a). Relevant tendencies have been maintained in terms of metallic electronic structure and interface charge transfer (Supplementary Fig. 59b,c) as well as the Janus electronic state (Supplementary Fig. 60) in the expanded system. It also demonstrates the OER-active $\text{Ir}^{\text{interf}}$ sites (Supplementary Fig. 61-64, Supplementary Table 10) and similar $\text{Ir}^{\text{tip}} \rightarrow \text{Ir}^{\text{interf}} \rightarrow \text{NiPS}_3$ hydrogen spillover channels (Supplementary Fig. 65, Supplementary Table 11)... (1st paragraph, Page 13).

Newly added Supplementary Fig. 59. (a) The atomic model and (b) DOS of expanded Ir/NiPS₃. (c) Work functions of Ir NPs and NiPS₃.

Newly added Supplementary Fig. 60. The charge density difference between different parts in expanded Ir/NiPS₃ model by Bader charge analysis, showing a clear Janus electronic state.

Newly added Supplementary Fig. 61. Energy barriers for four-electron-step OER process in the expanded Ir/NiPS₃ model. The Ir^{interf} site exhibits the lowest energy barrier, revealing its high activity.

Newly added Supplementary Fig. 62. Atomic models of the expanded Ir/NiPS₃ model adsorbed (a) *OH, (b) *O, and (c) *OOH on Ir site.

Newly added Supplementary Fig. 63. Atomic models of the expanded Ir/NiPS₃ model adsorbed (a) *OH, (b) *O, and (c) *OOH on P site.

Newly added Supplementary Fig. 64. Atomic models of the expanded Ir/NiPS₃ model adsorbed (a) *OH, (b) *O, and (c) *OOH on S site.

Newly added Supplementary Fig. 65. Calculated free energy diagram for hydrogen spillover on the expanded Ir/NiPS₃ model.

[Comment 3]. We can't use the work function of the Ir(111) surface to represent that of the nanoparticles. They may be much different.

[Response] Thanks for your valuable comments. As you pointed out, the bulk phase work function of Ir (111) cannot represent that of Ir nanoparticles. Upon careful examination of the computational data, the work function value used in the manuscript were indeed obtained from the calculation of Ir nanoparticles. The confused expressions and Fig. 1d have been modified and updated to make the manuscript more precise.

<After revision>

...the work function difference ($\Delta\psi$) between Ir nanoparticle and NiPS₃ support was evaluated in Fig. 1d. It indicates an overall electron transfer from Ir... (2nd paragraph, Page 4).

Updated Fig. 1d

[Comment 4]. When the in-situ Raman spectra were discussed, the authors mentioned that “the peaks of $E_g^{(2)}$ undergoes a blue shift during the HER”, and “this blue shift may be due to hydrogen spillover from Ir NPs to NiPS₃”. I agree with the authors, the blue shift may be used as an indicative descriptor. However, it is not direct evidence of the hydrogen spillover. This is because the authors cited Ref. 30 to support this key discovery of this project. Ref. 30 didn't clearly mention that the adsorption of H can lead to the blue shift, as the authors suggested in Fig. S37. So, I don't think the in-situ Raman observations have confirmed hydrogen spillover phenomenon and tip enhancement effect for water adsorption and splitting”.

[Response] Thanks for your pointing out the incorrect cited reference to improve the manuscript more precise. Indeed, the original Ref. 30 (Adv. Funct. Mater. 2022, 32, 2112750) is unrelated to the evidence for hydrogen spillover. In the revised manuscript, we have checked and updated the relevant references carefully to fit the discussions. We have re-organized the text and references to make the Raman-hydrogen spillover point more clearly.

The new reference has been updated to Ref. 32 (Sci. Rep. 2016, 6, 20904) to understand the different vibrational modes of NiPS₃, which we have cited to demonstrate the sensitivity of the $E_g^{(2)}$ in-plane stretching vibration mode (Fig. S37) to surface structural changes in NiPS₃. However, it is insufficient to solely attribute the blue shift of $E_g^{(2)}$ to hydrogen spillover only based on Ref. 32. Therefore, we have also referenced Ref. 8 (Angew. Chem. Int. Ed. 2021, 133, 16758-16763) in the manuscript. In Ref. 8, the peak of the $E_g^{(1)}$ vibration mode, associated with the symmetric stretching of O-Ti-O, undergoes a blue shift after applying an electric potential. This vibration mode, similar to our $E_g^{(2)}$, is sensitive to in-plane stretching. The authors reasonably speculate that the blue shift is attributed to hydrogen spillover from Pt NC to the TiO₂ support. Therefore, we have combined the two references to make the judgment that the blue shift indicates hydrogen spillover.

Supplementary Figure S5. Schematic representation (top view, side view) of vibrational amplitudes of $M_2P_2S_6$ unit cell atoms, for the five in-plane E_g and three out-of-plane A_g phonon modes.²⁰ Covalent bonds within $(P_2S_6)^{4-}$ anions (black lines) are indicated.

New Ref. 32 Sci. Rep. 6, 20904 (2016)

Figure 4. a) Optimized geometric structures of Pt NCs anchored on TiO₂ supports with/without oxygen vacancies (V_{O} -Pt/TiO₂ and Pt/TiO₂) and corresponding DFT calculation of the free energy. b) In situ Raman spectra of V_2O_5 -rich Pt/TiO₂ at -0.1 V vs. RHE with different electrolysis durations. c) Magnified view of the E_{g21} mode. d-f) In situ Raman spectra of V_2O_5 -rich Pt/TiO₂ (d), V_2O_5 -deficient Pt/TiO₂ (e), and TiO₂ (f) at a potential of -0.2 V vs. RHE.

Ref.8 Angew. Chem. Int. Ed. 133, 16758-16763 (2021)

Additionally, in-situ Raman spectroscopy demonstrates the generation of H_3O^+ and the creation of a locally acidic environment in the Ir/NiPS₃ system (Fig. 5e). Combined with theoretical analysis (Fig. 1b), it is reasonable to infer that the presence of Ir enhances the system's ability to undergo water dissociation and produce protons (*Nat. Commun.* 2022, 13, 2024). Supplementary Fig. 34 demonstrates the enhanced hydrophilicity and water adsorption capacity of Ir/NiPS₃ relative to NiPS₃.

Importantly, following your suggestion, we have also referred to additional literatures and incorporated them into our work to supplement the evidence for the hydrogen spillover and the water adsorption/dissociation enhancement:

i) The visual confirmation of the existence of hydrogen spillover has been achieved through the color change observed when mixing the Ir/NiPS₃ catalyst with WO₃ (Supplementary Fig. 38). In an alkaline environment, the color of WO₃ remained unchanged after HER testing. However, the mixture of Ir/NiPS₃ and WO₃ exhibited a deep blue color after HER testing, indicating the occurrence of hydrogen spillover, as the spillover hydrogen migrated and readily reacted with WO₃ to form the deep blue H_xWO₃ (*Nat. Commun.* 2020, 11, 4773; *Nat. Commun.* 2022, 13, 1189).

Newly added Supplementary Fig. 38. Color change photographs of (a) WO₃, (b) WO₃ after HER, (c) Ir/NiPS₃-WO₃ mixture, and (d) Ir/NiPS₃-WO₃ mixture after HER....

ii) The catalytic systems' hydrogen adsorption and desorption kinetics were analyzed by in situ CV to support the occurrence of hydrogen spillover (*Nat. Commun.* 2021, 12, 3502). Hydrogen adsorption and desorption peaks were monitored for Ir/NiPS₃, NiPS₃, Ir NPs, and the typical hydrogen spillover material Pt/WO₃ (Supplementary Fig. 39). NiPS₃ showed no clear adsorption and desorption peaks, indicating its weaker hydrogen adsorption capability. Supplementary Fig. 40 shows the corresponding relationship between the hydrogen adsorption peak position (HAPP) and the hydrogen desorption peak position (HDPP) with the scan rate. It

is demonstrated that Ir/NiPS₃ exhibited a lower absolute slope value similar to the typical Pt/WO₃ system, indicating accelerated hydrogen adsorption and desorption kinetics. This accelerated hydrogen adsorption/desorption kinetics may stem from the efficient hydrogen spillover effect (*Angew. Chem. Int. Ed.* 2019, 58, 16038-16042; *Nat. Commun.* 2021, 12, 3502).

Newly added Supplementary Fig. 39. CV profiles of (a) Ir/NiPS₃, (b) EE NiPS₃, (c) Pt/WO₃, (d) Ir NPs, (e) Pt/C and (f) Ir/C catalysts ...weaker hydrogen adsorption capability.

Newly added Supplementary Fig. 40. Plots of (a) hydrogen adsorption peak position and (b) hydrogen desorption peak position...This accelerated hydrogen adsorption/desorption kinetics may stem from the efficient hydrogen spillover effect.

iii) In addition to hydrogen adsorption and desorption, *in situ* Raman spectroscopy provides evidence of hydrogen migration (Fig. 5f). Furthermore, the hydrogen/deuterium (H/D) kinetic isotope effects (KIEs) can reflect the kinetic information of hydrogen or proton transfer in chemical reactions. Compared to the polarization curve of Ir/NiPS₃ in 1 M KOH/H₂O solution, the current density in 1 M KOD/D₂O is significantly reduced by approximately 2.7~4.6 times across the entire potential range (KIEs = 2.7~4.6) (Supplementary Fig. 41). The presence of KIEs (KIEs > 1.5) is considered as evidence that proton or hydrogen transfer is a key factor affecting reaction rate, confirming the possible existence of hydrogen spillover (*Nat. Commun.* 2019, 10, 5074; *ACS Catal.* 2021, 11, 68-73).

Newly added Supplementary Fig. 41. Polarization curves of Ir/NiPS₃ catalyst in aqueous 1 M KOH/H₂O solution and 1 M KOD/D₂O solution...

iv) The presence of a single-phase angle in the Bode plot indicates that the HER of the sample follows the Volmer-Tafel mechanism (*Adv. Funct. Mater.* 2023, 2301343). The only phase angle in the Supplementary Fig. 42a,b represent the Volmer process (water dissociation, $*+H_2O+e^- = *H+OH^-$). The slope of Ir/NiPS₃ (0.235) is greater than pure NiPS₃ (0.190), demonstrating the enhanced dynamics of water dissociation step relative to NiPS₃ (Supplementary Fig. 42c). Furthermore, KSCN poisoning experiments for metal sites have been conducted in Supplementary Figs. 43-44. It is found that the current densities of Ir/NiPS₃ and Pt/WO₃ were significantly reduced together with a rapid increase in the phase angle, which indicates that the inhibition of the Ir and Pt active sites reduces the kinetics of water dissociation. Combining with the evidence from Fig. 5e, h, j, and Supplementary Fig. 34, it is jointly revealed that the tip enhancement effect of Ir nanoparticles promotes water adsorption and splitting.

Newly added Supplementary Fig. 42. The Bode plots at different HER overpotentials of (a) Ir/NiPS₃ and (b) NiPS₃ catalysts obtained by *operando* EIS testing. (c) The plot of phase angle peak values vs. HER overpotentials.....

Newly added Supplementary Fig. 43. Catalyst poisoning experiment for Ir/NiPS₃. (a) LSV plot and (b) Bode plot at an overpotential of -100 mV vs. RHE...

Newly added Supplementary Fig. 44. KSCN poisoning experiments for the Pt/WO₃ catalyst. (a) LSV plot and (b) Bode plot at an overpotential of -100 mV vs. RHE.

< **Revisions in the text** > ...Note that the $E_g^{(2)}$ in-plane tensile vibration mode of NiPS₃ (*ca.* 177.1 cm⁻¹, Supplementary Fig. 37a) is very sensitive to changes in the surface structure of NiPS₃³². As for the Ir/NiPS₃ sample (Fig. 5f), ... (1st paragraph, Page 12).

32. Kuo, C-T. et al. Exfoliation and Raman spectroscopic fingerprint of few-layer NiPS₃ van der Waals crystals. *Sci. Rep.* **6**, 20904 (2016).

...According to previous protocols⁸, it is reasonable to speculate that this blue shift may be due to hydrogen spillover from Ir NPs to NiPS₃. Moreover, Supplementary Fig.38-44 provide the additional evidence for alkaline hydrogen spillover and the promoted water dissociation kinetic by the loaded Ir^{9,33-36}, details are presented in the Supplementary Information... (1st paragraph, Page 12).

...potentials corresponding to different current densities on the same electrode pair. The *in situ* CV and *operando* EIS measurements were performed at specified scan rates and overpotentials to verify the occurrence of hydrogen spillover and enhanced water dissociation kinetics, respectively... (1st paragraph, Page 17).

[Comment 5]. *The results from Ref. 30 demonstrated that the consideration of the magnetic structure of NiPS₃ is important. It is unclear whether the authors considered the magnetic structure of NiPS₃ in their DFT computations.*

[Response] Thanks for your valuable comments. At the beginning of our theoretical simulations, we did consider the impact of the magnetic properties of NiPS₃; but ultimately, we reasonably conducted to perform calculations in a system without considering spin. The reasons are as follows:

i) Similar to the description in Ref. 30 in the original manuscript, NiPS₃ exhibits a stable antiferromagnetic structure to minimize energy. The electron density of the up and down spins within the NiPS₃ system cancels each other out, resulting in a net magnetic moment of 0 (**Fig. R3**). Therefore, NiPS₃ is not a ferromagnetic material like Fe, Co, or Ni simple substance. Furthermore, Ir metal itself is non-magnetic, so the combination of the two materials almost cannot result in a ferromagnetic system.

ii) Even considering the changes in spin density due to the distortion and defects of the composite system, the net magnetic moment generated is extremely small (**Fig. R4a**). The spin density still only resides at the basal Ni atoms, which has little effect on the evolution of the adsorbate at the basal plane and Ir sites. Additionally, the consideration of spin does not alter the density of states (DOS) and the Janus electronic state effect of the heterostructure (Fig.

R4b,c). Furthermore, the thermodynamic calculation results of specific sites only differ at the Gibbs free energy values from those in the manuscript, and their trends and correct conclusions are not affected (**Fig. R5-R6**).

iii) The Vibrating Sample Magnetometer (VSM) experiment was used to test the Ir/NiPS₃ sample, and the results showed insignificant hysteresis loops and very small slopes, similar to NiPS₃, indicating the non-ferromagnetic characteristics of the Ir/NiPS₃ system (**Fig. R7**).

Fig. R3. (a) The spin density pattern and calculation result, and (b) DOS of NiPS₃ when considering electron spin.

Fig. R4. (a) The spin density pattern and calculation result, (b) DOS, and Bader charge analysis of Ir/NiPS₃ when considering electron spin.

Fig. R5. (a) Energy barriers for HER and (b) OER processes when considering electron spin.

Fig. R6. Calculated free energy diagram for hydrogen spillover on Ir/NiPS₃ when considering electron spin.

Fig. R7. Hysteresis curve of NiPS₃ and Ir/NiPS₃.

[Comment 6]. Fig. 5l shows the free energy diagram to support the hydrogen spillover process. It is not enough. The energy barrier for each step is needed.

[Response] Thanks for your valuable comments. According to the reviewer's suggestion, we used the climbing image nudged elastic band method (CINEB) to calculate the energy barriers at each step, as shown in the updated Fig. 5l. The results indicate that the hydrogen transitions between the remaining steps are smooth, with only small energy barriers present at the interface and between the two adsorption sites on the substrate (0.16-0.09=0.07 eV, 0.49-0.14=0.35 eV, and 1.28-0.41=0.87 eV respectively). Compared to previous work (*Nat. Commun.* 2021, 12, 3502; *Nat. Commun.* 2022, 13, 5382), such small energy barriers are easily overcome, and the hydrogen spillover effect in the Ir/NiPS₃ system remains thermodynamically reasonable. These statements have been reflected in the revised manuscript.

<After revision>

...and its value at the NiPS₃ substrate is sufficient to be greater than 0, resulting in the H* desorption tendency on the substrate. Meanwhile, low proton spillover energy barriers (0.07 eV,

0.35 eV, 0.87 eV) were observed at the interface and substrate, which will be easy to overcome according to previous reports^{9, 40}. From the thermodynamic point, ... (1st paragraph, Page 13). ... Charge density difference studies were employed to analyze electron transfer within the system⁴⁷. The clipping image nudged elastic band method (CINEB) was used to calculate energy barriers⁴⁸. For finite element simulation, the Gouy-Chapman model's Nernst-Planck-Poisson... (1st paragraph, Page 18).

Updated Fig. 51

[Responses to reviewer 2's comments]

[Comment]. *Dai et al. developed a Ir clusters loaded NiPS₃ metal-support structure, and applied it as an efficient bifunctional HER and OER catalyst. In alkaline environments, it shows a low η_{10} potential of 23 mV for HER and excellent OER/UOR potentials. I am very impressive with the finding of Janus electronic state for the supported Ir nanoclusters as well as the roles in activating the hydrogen spillover and water oxidation. In-situ observations indicated the local acidic environment and the dynamic hydrogen spillover in HER, and found the anti-reconstruction merit in OER. The degradation of urea by UOR is also interesting. Overall, this work presents the novel and essential point of Janus electronic state for metal-support structures, and justifies their findings with both theoretical and experimental evidences. Hence, I recommend this paper to be published on Nat. Commun. after carrying out a minor revision to figure out the following points.*

[Response] We appreciate the valuable comments from the reviewers. Indeed, the reviewers' advice and recommendations were very prompt and helpful. We revised the manuscript according to the reviewers' directions. Thank you very much.

[Comment 1]. *The wording in some parts is not clear and makes people confused. For example, the description of the ligand effect of ethylene glycol is not detailed enough, whether ethylene glycol is unintentionally introduced or necessary in synthesis, and whether other polyols can synthesize Ir nanoparticle loaded materials. Besides, the influence of ethylene glycol ligands on Janus electronic states needs further elaboration.*

[Response] Thanks for your concern on this aspect. We would like to offer the following explanations for your inquiries: Firstly, regarding the study on the promotion of hydrogen spillover effect by ethylene glycol ligands, detailed descriptions have been previously provided in the literature (*Energy Environ. Sci.* 2019, 12, 2298-2304). The ethylene glycol ligands are believed to enrich the hydrogen concentration around Pt and promote hydrogen spillover. In our work, ethylene glycol ligands also play a role in providing electrons to the top Ir atoms to promote proton adsorption, and their electron-donating function also consolidates the Janus electronic state. Some relevant expressions have been added in the revised manuscript.

Secondly, the ethylene glycol reduction method is a common approach for synthesizing metal-carrier materials, and the introduction of ethylene glycol is a necessary step in the synthesis process, as confirmed in Supplementary Fig. 9. Fortunately, the ethylene glycol ligands also play a positive role in the formation of the Janus electronic state. Lastly, other

polyols can also be used to synthesize Ir nanoparticle-loaded materials, for instance, the use of diethylene glycol ligands has similarly led to the synthesis of Ir nanoparticle-loaded NiPS₃ materials (Fig. R8), demonstrating the potential of the polyol reduction method.

<After revision>

... This bidirectional electron transfer manner induces the formation of e^- -deficient Ir^{interf} region and e^- -rich Ir^{tip} region, verifying the hypothesis of Janus electronic state in Fig. 1b. In addition, the EG ligand supplies electrons to Ir^{tip} to consolidate this Janus electronic state, which is beneficial for proton enrichment at the tip. We further indicate the plane average potential along the X direction... (3rd paragraph, Page 4).

Fig. R8. (a, b) TEM image, (c) SAED pattern, (d) HR-TEM image, and (e) EDS mapping of Ir/NiPS₃ synthesized by diethylene glycol reduction method.

[Comment 2]. *Have the authors considered the single atom effect, such as loading Ir as a single atom instead of a particle? Current reports on single atom loading of Ir (Nat. Commun., 2023, 14 (1): 2588; Adv. Material, 2021, 33 (8): 2007056.) all demonstrate their unique advantages. Please provide more explanation.*

[Response] As you kindly pointed out, single-atom catalysts have been a hot topic of research in recent years, owing to their exceptionally high intrinsic activity, atom utilization efficiency, and appealing cost-effectiveness in the use of precious metals, making them promising new catalytic materials. The aggregation effect of metal atoms into single atoms and nanoparticles endows each with unique advantages. The debate in the catalysis field regarding the performance merits of the two is not characterized by a single attitude (Nat. Catal. 2022, 5, 485-493). Therefore, we believe that rational selection of single atoms or nanoparticles based on the specific environment and requirements is essential to maximize their effectiveness.

In our work, the essence of hydrogen spillover requires the adsorption and desorption of hydrogen to occur in two different locations, with the production of protons in basic environments contingent on water dissociation. Based on numerous prior literature reports related to hydrogen spillover (*Nat. Commun.* 2021, 12, 3502; *Nat. Commun.* 2022, 13, 5382), we believe that nanoparticles are more favorable for the construction of hydrogen spillover catalytic systems. The numerous active sites of nanoparticles fulfill the requirements for water dissociation and proton accumulation, serving as proton activation centers, which is very crucial. Furthermore, based on the earlier analysis of the Janus electronic state, the interface contact between the particles and the substrate leads to the generation of electron-deficient centers, which also facilitates the OER reaction. Therefore, considering these factors comprehensively, Ir nanoparticles was taken as the primary choice for this work.

[Comment 3]. In Figure 3d, from the XAFS date, the Ir exhibits a super high valence state that surpasses Ir in IrO₂. How can understand this? Will it also occur in other metal-support structure? If it is, the authors need to check and provide reference.

[Response] Thanks for your concern on this aspect. The white line intensity of the L₃-edge XANES is related to the density of unoccupied d orbitals. After being supported on the substrate, the strong ionicity and Oh symmetry around the Ir cation will lead to an enhancement in the white line intensity. An increase in the white line peak intensity in supported metals over corresponding oxides can be occurred, for example, in Ir NP-MgO, Ir-MgO, Pt-MgO, etc (*J. Phys. Chem.* 1994, 98, 6258-6262; *Catal. Surv. Asia* 2023, 27, 95-106). Additionally, Ir can also exhibit higher oxidation states than IrO₂ (*Adv. Mater.* 2020, 32, 2000872).

Fig. 4 Ir L₃-edge XANES profiles of Ir-NP/MgO treated at 1073 K and reference compounds (Ir⁰ powder, Ir₂O₃, and IrO₂)

Figure S4. XANES of solid solutions of transition metal elements and MgO, and oxides. (a) Ir L₃-edge, (b) Pt L₃-edge.

Catal. Surv. Asia, 2023, 27(1): 95-106

ACS Catal. 2021, 11, 4, 1952–1961, Figure 3a

Adv. Mater., 2020, 32(24): 2000872, Figure 4c

[Comment 4]. Please comment on the performance comparison or active sites difference compared to some reports of Ir cluster or particles like 10.1016/j.jechem.2023.08.017 or DOI: 10.26599/NRE.2023.9120056.

[Response] Thanks for your valuable comment. We have compared and commented on the reports of these Ir clusters and particles in the revised manuscript.

<After revision>

...(UA-OWS, Fig. 4g inset) with good durability (Supplementary Fig. 32). The OWS and UA-OWS performances are comparable to many current excellent catalytic systems (Supplementary Table 4,5)²⁴⁻²⁷. It is also found that the catalytic electrode pair could remain stable for over 1000 hours... (3rd paragraph, Page 10).

26. Wang, C., Yu, L., Yang, F & Feng, L. MoS₂ nanoflowers coupled with ultrafine Ir nanoparticles for efficient acid overall water splitting reaction. *J. Energy Chem.* **87**, 144-152 (2023).

27. Wang, C., Schechter, A & Feng, L. Iridium-based catalysts for oxygen evolution reaction in acidic media: Mechanism, catalytic promotion effects and recent progress. *Nano Res. Energy* **2**, e9120056 (2023).

[Comment 5]. Although this work focuses on the alkaline performance, it still remain some parts lack of detailed interpretation or descriptions. For example, in Figure S16, compared to alkaline environments, the performance of the Ir/NiPS₃ heterostructure on HER in acidic media is slightly worse, lower than commercial Pt/C. How to understand this phenomenon?

[Response] Thanks very much for your comment. Compared with alkaline environment, the slightly inferior HER performance in acidic medium is because 2D NiPS₃ materials are always not inherently tolerant to acidic environments. At present, most of the tests on the catalytic performance of NiPS₃-based materials are done in an alkaline environment (*Small* 2019, 15, 1902427, *Adv. Funct. Mater.* 2020, 30, 1908708; *ACS Nano* 2018, 12, 5297). Furthermore, it is widely acknowledged that the performance of Pt/C catalysts in acidic environments is nearly optimal; however, their sluggish HER kinetics in alkaline conditions due to following the Volmer-Heyrovsky mechanism has been noted (*Adv. Funct. Mater.* 2023, 2301343). In contrast, Ir/NiPS₃, which follows the Volmer-Tafel mechanism, exhibits superior kinetic rates (Supplementary Fig. 42a). In summary, Ir/NiPS₃ is more inclined to HER electrocatalysis in alkaline environments. Its low corrosion maintains structural integrity, and the concurrent promotion of the hydrogen spillover effect contributes to its advanced performance.

<After revision>

Newly added Supplementary Fig. 42a

[Comment 6]. Transition metal sites are generally considered active sites for catalytic reactions due to the tunability of *d* electrons. However, in the DFT calculation section, the authors only calculated P, S, Ir sites, but did not consider Ni sites. Please provide a detailed explanation.

[Response] Thanks for your valuable comments. In the DFT calculations, the screening for an active site is based on whether it can adsorb intermediates and its adsorption energy value. For the NiPS₃ substrate, the Ni sites are not exposed according to the NiPS₃ structural feature, so the intermediates are hard to adsorb on the Ni sites. In this regard, the catalytic process mainly takes place at P and S sites on the surface (*Small* 2019, 15, 1902427). In addition, the key focus of our work is to simulate the hydrogen spillover effect between Ir nanoparticles and NiPS₃, as well as the OER at the interface. Therefore, it does not involve the Ni atoms located internally.

[Comment 7]. I still wonder why the adsorption energy of the Ir site in Figure 5k is lower after the formation of heterostructures (from -0.26 eV to -0.47 eV), but the authors indicated the heterostructure is more favorable in HER. As we know, ΔG closer to 0 corresponds to more favorable HER. How can this confusion be understood?

[Response] Thanks for your valuable comments. In general, ΔG_{H^*} can be served as an indicator for evaluating the HER performance of individual active sites, where a ΔG_{H^*} value closer to 0 indicates a hydrogen adsorption/desorption capability approaching equilibrium, which is more favorable for HER. However, in the Ir/NiPS₃ system, the hydrogen spillover effect does not operate through a single-site mechanism. It requires Ir^{tip} to act as a proton concentration site,

while the NiPS₃ basal plane serves as a proton desorption site. Therefore, Ir^{tip} has a more negative ΔG_{H^*} value than pure Ir, and it is more beneficial for proton enrichment. Additionally, we note that the ΔG_{H^*} of the P sites in Ir/NiPS₃ is smaller than that of the P sites in NiPS₃, promoting the HER reaction. This comparison is based on the traditional single-site systems and does not present a contradiction. To avoid confusion, we have modified the relevant statements in the revised manuscript.

<After revision>

...Moreover, the ΔG_{H^*} values of individual sites in different systems are compared in Fig. 5k, Supplementary Fig. 56,57, and Supplementary Table 9. Relative to NiPS₃ (0.61 eV-P site),...
(1st paragraph, Page 13).

...make the ΔG_{H^*} even more negative^{38, 39}. This enhancement of proton concentration effect benefits the hydrogen spillover mechanism based on multiple sites. Specific reaction site analyses were further conducted...(1st paragraph, Page 13).

[Responses to reviewer 3's comments]

[Comment] *The paper by Liu et al. conducted the induction of Janus electronic states in an Ir nanoparticle-loaded NiPS₃ system to enhance alkaline hydrogen evolution and OER/UOR reactions. This study combines experimental characterization, catalytic testing, and DFT/COMSOL calculations to understand the reasons for forming Janus electronic states and the mechanisms that promote reactions. The concept and research content is interesting and well presented. Additionally, Ir/NiPS₃ bifunctional catalysts exhibit impressive performance, such as low 23 mV/236 mV under alkaline conditions η_{10} HER/OER potential, while also possessing UOR urea degradation ability. Therefore, I recommend its publication in Nat. Commun., after the following concerns and remaining issues are addressed appropriately:*

[Response] We appreciate the valuable comments from the reviewers. Indeed, the reviewers' advice and recommendations were very prompt and helpful. We revised the manuscript according to the reviewers' directions. Thank you very much.

[Comment 1]. *In Fig. 1, the authors chose to adsorb an ethylene glycol ligand while constructing the model, which raises doubts about whether the emergence of the Janus electronic state is related to the ethylene glycol ligand. In other words, if the system did not contain the ethylene glycol ligand, would the Janus electronic state not exist? This is fundamental to the discussion of the mechanism in the article and is also the starting point for the authors' choice of the ethylene glycol method for synthesizing the material. Please provide a reasonable explanation.*

[Response] Thanks for your concern on this aspect. When the theoretical analysis was initially conducted, we considered the presence of ethylene glycol ligands. This is because the ethylene glycol reduction method is more suitable for the loading of Ir nanoparticles, making the introduction of ligands inevitable. Of course, the ethylene glycol ligands are beneficial for hydrogen spillover, which is an important reason for their inclusion in our model construction (*Energy Environ. Sci.* 2019, 12, 2298-2304). However, the generation of the Janus electronic state is not attributed to the action of ethylene glycol, but is based on fundamental physical and chemical effects (Fig. 1b). The transfer of interface electrons and tip enhancement form the basis for the creation of the Janus electronic state. This is reflected in the Bader charge analysis results shown in Fig. 1e. The ethylene glycol ligands only play a supporting role, namely, supplying electrons to Ir^{tip} to strengthen the Janus electronic state, and are not the key factor in

its formation. In order to clarify the role of ethylene glycol, we have made relevant statements in the revised manuscript.

<After revision>

...verifying the hypothesis of Janus electronic state in Fig. 1b. In addition, the EG ligand supplies electrons to Ir^{tip} to consolidate this Janus electronic state, which is beneficial for proton enrichment at the tip. We further indicate the plane average potential... (3rd paragraph, Page 4).

[Comment 2]. *The authors need to declare whether the stability of 1000 hours in Figure 4h was tested on the same sample at different current densities, or was the sample replaced midway?*

[Response] Thanks for your concern on this aspect. The stability of 1000 hours in the manuscript was obtained through continuous testing of the same catalytic electrode pair at different current densities. The relevant statements are provided in the revised manuscript.

<After revision>

...Chronopotentiometry was employed to evaluate the long-term stability of HER, OER, and OWS for the samples. The 1000 hours chronopotentiometry measurement for OWS was operated at the corresponding potentials of different current densities on the same electrode pair... (1st paragraph, Page 17).

[Comment 3]. *Hydrogen spillover itself is an effect under acidic conditions. Since Ir/NiPS₃ has alkaline hydrogen spillover effect, it should also have acidic hydrogen spillover effect. According to Fig. S16 characterized by the authors, the acidic HER performance of the sample does not exceed that of Pt/C catalyst. Why is its performance not outstanding compared with that of alkaline HER?*

[Response] Thanks for your valuable comment. Currently, most tests evaluating the catalytic performance of NiPS₃-based materials are conducted in alkaline environments (*Nat. Commun.* 2023, 14, 6462; *Adv. Funct. Mater.* 2020, 301908708; *ACS Nano* 2018, 125297). This is because NiPS₃ exhibits stronger intrinsic activity and stability in alkaline media (Fig. 4a, Supplementary Fig. 16). This may be due to the poor acid tolerance and corrosion susceptibility of NiPS₃-based materials themselves. Furthermore, the performance and kinetics of Pt/C catalysts in acidic environments are nearly optimal, making it challenging to surpass these levels. In contrast, Pt/C follows the Volmer-Heyrovsky mechanism in alkaline conditions, showing slow HER kinetics (*Adv. Funct. Mater.* 2023, 2301343). This allows the optimized hydrogen spillover effect in Ir/NiPS₃ to surpass the performance of Pt/C. In summary, NiPS₃ is

more inclined towards HER electrocatalysis in alkaline environments, which is also the starting point for our construction of Ir/NiPS₃ and the study of its alkaline hydrogen spillover mechanism.

[Comment 4]. *The potential drop usually occurs due to the solution resistance and needs to be eliminated the catalytic properties. Were the potential drops compensated for the electrocatalytic performance testing?*

[Response] Thanks for your valuable comment. The potential drops have all been compensated in our study. To clarify this, we have added the related descriptions in the revised manuscript.

<After revision>

...underwent 20 cycles of cyclic voltammetry (CV) at a scan rate of 100 mV s⁻¹. The data were further used for the Ohmic drop (iR) correction. Charge transfer resistance (R_{ct}) was fitted using... (1st paragraph, Page 17).

[Comment 5]. *The Supplementary Notes points out that $\Delta G_{(H^*)} = \Delta E_{(H^*)} + 0.24\text{eV}$. However, the data in Supplementary Table 9 seems not to be the case. Please check carefully.*

[Response] Thanks for your valuable comment. Due to our later consideration that the adsorbate frequency on the particle and NiPS₃ substrate will be different, we make separate correction of the Gibbs free energy for the proton adsorbed at each site at 298.15 K, rather than using a uniform correction standard of 0.24 eV. However, we neglected to modify the relevant content in the original Supplementary Note 2. These issues have been rectified in the revised manuscript.

<After revision>

...Furthermore, the Gibbs free energies for hydrogen adsorption in the HER process were corrected at 298.15 K, ΔG_{H^*} can be evaluated by Equation 17:

$$\Delta G_{H^*} = \Delta E_{H^*} + \Delta E_{ZPE} - T\Delta S_H \quad (17)$$

where ΔE_{H^*} , ΔE_{ZPE} and $T\Delta S_H$ represent the differences of hydrogen adsorption energy, zero-point energy, and the entropy between adsorbed hydrogen and hydrogen in the gas phase, respectively. The calculation results are shown in... (Supplementary Information, Page S3).

[Comment 6]. *For OER reaction, the authors seem to have considered the pH value of the reaction (SI, formula 5-8), and the theoretical potential has been subtracted by 0.401 instead*

of 1.23 V, indicating that the role of pH value has been considered again. Is this method of considering pH reasonable?

[Response] Thanks for your valuable comment. In the Gibbs free energy equations and the calculation of overpotential, we have considered the pH value. This method of reflecting pH is correct and has been fully reflected in many published and newly published papers (*Nat. Commun.* 2018, 9, 3376; *Angew. Chem. Int. Ed.* 2023, 135, e202214259).

i) For Gibbs free energy equations: As long as the reaction involves $[H^+]$ or $[OH^-]$, pH will affect the free energy of the reaction. The reaction free energy of $[H^+]$ and $[OH^-]$ of different concentrations relative to the standard hydrogen electrode is as follows:

$$H_3O^+ + e^- = \frac{G_{H_2}}{2} + H_2O \quad \Delta G = 0.0592 \times pH \text{ eV} \quad (R1)$$

$$H_2O + e^- = \frac{G_{H_2}}{2} + OH^- \quad \Delta G = 0.0592 \times pH \text{ eV} \quad (R2)$$

$$\Delta G = \Delta G^0 - K_B T \ln \left(\frac{1}{[H^+]} \right) = 0.0592 \times pH \quad (R3)$$

Therefore, when $pH \neq 0$,

$$G_{H^+} + G_{e^-} = \frac{G_{H_2}}{2} - 0.0592 \times pH \quad (R4)$$

Besides, the four-electron step reaction of OER can be written as Equation S1-S4 (Supplementary Note 1). Using $\frac{G_{H_2}}{2} - 0.0592 \times pH$ to evaluate $G_{H^+} + G_{e^-}$, the Gibbs free energy variations of the four reaction steps are represented as ΔG_1 , ΔG_2 , ΔG_3 , and ΔG_4 , which can be calculated as:

$$\Delta G_1 = G_{*OH} + \left(\frac{G_{H_2}}{2} - 0.0592PH \right) + eU - G_* - G_{H_2O} \quad (R5)$$

$$\Delta G_2 = G_{*O} + \left(\frac{G_{H_2}}{2} - 0.0592PH \right) + eU - G_{*OH} \quad (R6)$$

$$\Delta G_3 = G_{*OOH} + \left(\frac{G_{H_2}}{2} - 0.0592PH \right) + eU - G_{*O} - G_{H_2O} \quad (R7)$$

$$\Delta G_4 = G_{O_2} + G_* + \left(\frac{G_{H_2}}{2} - 0.0592PH \right) + eU - G_{*OOH} \quad (R8)$$

ii) For overpotential: $2H_2O = 4H^+ + 4e + O_2$, the standard electrode potential is calculated as: $V = \left[G_{O_2} - 2G_{H_2O} + 4 \times \left(\frac{G_{H_2}}{2} - 0.0592 \times pH \right) \right] / 4$. Considering the pH value, the V value is calculated as 0.401 eV at $pH = 14$ under alkaline conditions.

The Gibbs free energy variation of the rate-determining step can be described as:

$$\Delta G_{det.} = \max(\Delta G_1, \Delta G_2, \Delta G_3, \Delta G_4) \quad (R9)$$

The theoretical calculated overpotential η is:

$$\eta = \Delta G_{det.} - 0.401 \text{ for alkaline electrolyte} \quad (R10)$$

In general, our method of considering pH value is correct and reliable.

[Comment 7]. In Fig. 5k, why do the authors think that the performance of the P and S sites in Ir/NiPS₃ is improved compared to the pure material, while the performance of pure Ir is not as good as the Ir site in Ir/NiPS₃, even though $\Delta G_{(H^*)}$ is closer to 0 in both cases?

[Response] Thanks for your valuable comment. The improvement in the performance of P/S and Ir sites should be understood from different perspectives. In our manuscript, we mentioned that the enhancement of the performance of the P sites in the heterostructure is based on the traditional single-site mechanism. We used ΔG_{H^*} as an indicator to evaluate the HER performance of individual active sites, where a ΔG_{H^*} value close to 0 indicates a hydrogen adsorption/desorption capability approaching equilibrium, which is more favorable for the HER. However, in the Ir/NiPS₃ system, the hydrogen spillover effect does not operate through a single-site mechanism. It requires Ir^{tip} to act as a proton concentration site, while the NiPS₃ basal plane serves as a proton desorption site. Therefore, the ΔG_{H^*} value of Ir^{tip} is more negative than pure Ir, which is precisely required for enhancing proton enrichment and the hydrogen spillover. Thus, these two perspectives are not contradictory. To avoid this confusion, we have modified the relevant statements in the revised manuscript.

<After revision>

...Moreover, the ΔG_{H^*} values of individual sites in different systems are compared in Fig. 5k, Supplementary Fig. 56,57, and Supplementary Table 9... (1st paragraph, Page 13).

...their d-band centers to make the ΔG_{H^*} even more negative^{34,35}. This enhancement of proton concentration effect benefits the hydrogen spillover mechanism based on multiple sites.

Specific reaction site analyses were further conducted to... (1st paragraph, Page 13).

REVIEWERS' COMMENTS

Reviewer #1 (Remarks to the Author):

The authors have carefully addressed all the issues I raised. I, therefore, recommend it for publication.

Reviewer #2 (Remarks to the Author):

a good revision. The work can be published.

Reviewer #3 (Remarks to the Author):

The authors have well revised the manuscript, I recommend to accept it now.